# The head mesodermal cell couples FMRFamide neuropeptide signaling with rhythmic muscle contraction in *C. elegans*

Ukjin Choi [1,2,5], Mingxi Hu [2,5], Qixin Zhang[3] & Derek Sieburth [2,4] ✉

FMRFamides are evolutionarily conserved neuropeptides that play critical roles in behavior, energy balance, and reproduction. Here, we show that FMRFamide signaling from the nervous system is critical for the rhythmic activation of a single cell of previously unknown function, the head mesodermal cell (hmc) in *C. elegans*. Behavioral, calcium imaging, and genetic studies reveal that release of the FLP-22 neuropeptide from the AVL neuron in response to pacemaker signaling activates hmc every 50 s through an *frpr-17* G protein-coupled receptor (GPCR) and a protein kinase A signaling cascade in hmc. hmc activation results in muscle contraction through coupling by gap junctions composed of UNC-9/Innexin. hmc activation is inhibited by the neuronal release of a second FMRFamide-like neuropeptide, FLP-9, which functions through its GPCR, *frpr-21*, in hmc. This study reveals a function for two opposing FMRFamide signaling pathways in controlling the rhythmic activation of a target cell through volume transmission.

Neuropeptides represent the largest and most diverse class of neurotransmitters in the brain, and neuropeptide signaling plays critical roles in many behavioral processes, including feeding, memory, and sleep[1–3]. Disruption in neuropeptide signaling is implicated in numerous behavioral disorders[4–6]. Neuropeptides are commonly present as co-transmitters, but unlike fast neurotransmitters, neuropeptides act at slower timescales, can act over long distances, and have longer lasting effects on target cell activity[7,8]. Neuropeptide signaling in the brain has been typically regarded as modulatory, whereby neuropeptides positively or negatively regulate postsynaptic responses elicited by fast neurotransmitters[9]. However, neuropeptides can also function as excitatory transmitters in their own right by directly activating target cells. For example, in circuits regulating wakefulness, orexin secreted from the lateral hypothalamus generates robust postsynaptic spike trains in target neurons that are independent of those produced by co-released glutamate[10]. Pulse-generating KDNy cells in the hypothalamus control episodic activation of target neurons through the release of the FMRFamide-like neuropeptide kisspeptin[11]. The

neuropeptide FMRFamide elicits fast depolarizing inward currents in the snail nervous system[12]. Finally, in *C. elegans,* the neuropeptide-like protein NLP-40 depolarizes a pair of GABAergic motor neurons to control a rhythmic behavior[13,14].

The *C. elegans* defecation motor program (DMP) is a simple rhythmic behavior with a period of about 50 s that functions to expel digested food from the intestine and to regulate nutrient uptake[15,16]. Calcium oscillations in the intestine, which functions as the pacemaker[17,18], control the timing and execution of three sequential muscle contractions in the DMP starting with the posterior body wall muscle contraction (pBoc), followed by the anterior body wall muscle contraction (aBoc), and lastly, the enteric muscle contraction which leads to expulsion[16]. Prior studies have shown that neuropeptide signaling is essential for the proper execution of the aBoc step[14,19,20], but the underlying cellular and molecular mechanisms by which the pacemaker controls aBoc have not been defined.

The head mesodermal cell (hmc) is one of a small number of the 959 somatic *C. elegans* cells whose biological function has not yet been

[1]DSR graduate program, Keck School of Medicine, University of Southern California, Los Angeles, CA 90033, USA. [2]Zilkha Neurogenetic Institute, Keck School of Medicine, University of Southern California, Los Angeles, CA 90033, USA. [3]MPHY program, Keck School of Medicine, University of Southern California, Los Angeles, CA 90033, USA. [4]Department of Physiology and Neuroscience, Keck School of Medicine, University of Southern California, Los Angeles, CA 90033, USA. [5]These authors contributed equally: Ukjin Choi, Mingxi Hu. ✉e-mail: sieburth@usc.edu

determined experimentally. hmc is a prominent H-shaped cell situated in the head of the animal, that extends four processes anteriorly and posteriorly along both the dorsal and ventral side in the neck region of the animal[21,22]. The position, morphology, and transcriptional profile of hmc suggest that hmc may play a role in coupling intercellular neuropeptide signaling to neck muscle contraction. First, hmc is situated within the pseudocoelom, which is a fluid-filled cavity into which neuropeptides are secreted by surrounding neurons[19,23]. Second, serial section ultrastructural reconstructions show that the hmc cell body and posteriorly directed processes form numerous large gap junctions with body wall muscles along the neck[24], suggesting that hmc helps to coordinate the activity of the neck muscles. Third, transcriptomic analysis of hmc reveals a unique gene expression profile not shared by any other major tissue (skin, muscle, neuron, or intestine) that is highly enriched in a number of genes encoding neuropeptide GPCRs, as well as genes associated with GPCR and calcium signaling[25,26]. Finally, hmc is born late in embryogenesis, and misspecification of hmc cell fate does not cause lethality, implying a post-developmental role of hmc in mature animals[27–29].

Here, we identify a critical function for specific FMRFamide-like neuropeptides as volume transmitters to control the activation of hmc during the aBoc step of the DMP. We show that the motor neuron AVL rhythmically activates hmc every 50 s in phase with AVL activation and anterior body wall muscle contraction. We find the secretion of the neuropeptide-like protein FLP-22 from AVL is critical for hmc activation and neck muscle contraction. FLP-22 activates the FRPR-17 GPCR in hmc, which in turn controls calcium spike generation through a conserved PKA signaling pathway. Gap junctions composed of UNC-9/innexin subunits couple hmc activation with neck muscle contraction. The rhythmic activation of hmc is also directly inhibited by a second neuropeptide signaling pathway, originating from the nervous system, composed of the FLP-9 neuropeptide and its GPCR FRPR-21. Our study identifies a novel behavioral circuit in which the activation of hmc is controlled by both activating and inhibiting FMRFamide signaling to control a rhythmic behavior.

## Results

### The aBoc step is controlled by peptidergic signaling from the AVL neuron

The aBoc step consists of a contraction-relaxation of the neck muscles (Fig. 1a) that begins about three seconds after pBoc starts, and immediately before expulsion[16,30,31]. To investigate the role of AVL in aBoc, we genetically ablated AVL by expressing the Caspase-1/Interleukin-1 converting enzyme ICE under an AVL-specific promoter fragment ((Pnmur-3(1 kb)), Fig. S1a, b), used hereafter for all AVL-specific expression, and examined the frequency of the aBoc step during the DMP in freely moving animals under the dissecting microscope[32]. AVL ablation resulted in the aBoc step being absent in nearly all cycles, reducing aBoc frequency from 100% (once each cycle) to less than 10% (Fig. 1b), in agreement with laser ablation studies of AVL[30]. The neuropeptidelike protein, NLP-40, is the pacemaker timing signal whose release from the intestine evokes an action potential in two motor neurons, AVL and DVB, resulting in expulsion[13,14,33]. Mutants lacking either nlp-40 or its receptor, the GPCR aex-2 exhibited a significant decrease in aBoc frequency (Fig. 1b and ref. 34), albeit to a lesser extent compared to AVL ablation. The aBoc frequency defects of nlp-40 mutants could be fully rescued by expressing nlp-40 cDNA selectively in the intestine (using the ges-1 promoter, Fig. 1b). Similarly, the missing aBocs of aex-2 mutants could be restored by expressing aex-2 cDNA selectively in AVL (Fig. 1b). Thus, the pacemaker timing signal NLP-40 provides the major drive for aBoc through aex-2/GPCR signaling in AVL.

AVL is a GABAergic motor neuron that promotes the expulsion step through GABA release from neuromuscular junctions (NMJs)[31]. AVL does not promote aBoc via GABA release since unc-25/glutamate

decarboxylase null mutants, which lack GABA[16,35], have normal aBoc frequencies (Fig. S2a and ref. 16,30). AVL expresses a number of neuropeptide-like proteins as well as neuropeptide processing proteins[26]. Neuropeptides are secreted from dense core vesicles (DCVs) by SNARE-mediated exocytosis[36]. Expression of tetanus toxin (TeTx) selectively in AVL, which impairs SNARE-mediated exocytosis[37], reduced aBoc frequency to 24% (Fig. 1b). egl-3 encodes the ortholog of prohormone convertase 2 (PC2), which cleaves neuropeptide precursors in DCVs during maturation[38]. egl-3/PC2 mutants had mild expulsion frequencies (Fig. S2b), but had significantly reduced aBoc frequencies, and the aBoc defects could be rescued by expressing egl-3 cDNA selectively in AVL (Fig. S2a). These results indicate that neuropeptide released from AVL specifically controls the aBoc step.

### The FLP-22 FMRFamide-like neuropeptide positively regulates aBoc

To identify the neuropeptide(s) that control aBoc, we screened for genes encoding FMRFamides (FLPs) or neuropeptide-like proteins (NLPs) that reduce aBoc frequency when knocked down by RNA interference (RNAi). flp-22 encodes an FMRFamide neuropeptide protein with no previously reported function that is cleaved by egl-3/PC2[38], and is expressed at high levels in AVL. Knockdown of flp-22 by RNAi or knockout of flp-22 by deleting the flp-22 coding region significantly reduced aBoc frequencies to about 60% (Fig. 1c and S2a, c) without affecting DMP cycle lengths (Fig. S2d). flp-22 mutants exhibited Exp frequencies, locomotion rates, and egg laying rates that were similar to wild-type controls (Fig. S2b, e, f). Thus, flp-22 plays a specific role in promoting normal aBoc frequency. flp-22 is the second most abundant gene expressed in AVL, and flp-22 is more abundant in AVL than in any other cell[26]. The aBoc defects of flp-22 mutants could be fully rescued by expressing flp-22 cDNA in GABAergic motor neurons including AVL (using the unc-47 promoter), whereas expression of flp-22 cDNA in cholinergic neurons (using the unc-129 promoter) failed to rescue (Fig. 1c). Expression of flp-22 cDNA selectively in AVL (using nmur-3(1 kb) promoter) partially rescued the aBoc frequency defects of flp-22 mutants (Fig. 1c). Thus, flp-22 functions in AVL as well as one or more additional GABAergic neurons to control aBoc.

### The neuropeptide GPCR FRPR-17 functions in hmc to control aBoc

Neuropeptides exert their biological effects on target cells primarily through G protein coupled receptors. The FLP-22 pro-peptide is processed into three identical 9 amino acid-long peptides that have been detected in worm lysates[38]. To identify genes functioning downstream of FLP-22, we conducted a genetic screen for suppressors of the uncoordinated phenotype caused by flp-22 overexpression (flp-22 (OE)). One of the suppressors, uj249, carries a missense mutation in the extracellular domain of the orphan FMRFamide-like GPCR frpr-17 (Fig. S2c). uj249 mutants as well as frpr-17 null mutants, which carry a deletion of the entire frpr-17 coding sequence (Fig. S2c), had normal DMP cycle lengths and Exp frequencies (Fig. S2b, d), but had aBoc frequencies of 60% (Fig. 1c). Locomotion and egg laying rates of frpr-17 mutants were reduced compared to wild-type controls (Fig. S2e, f). Double mutants lacking both frpr-17 and flp-22 exhibited aBoc frequencies of 60%, which is similar to those of either flp-22 or frpr-17 single mutants (Fig. 1c). Thus, frpr-17 functions in a common genetic pathway with flp-22 to promote normal aBoc frequency.

To determine in which tissue frpr-17 functions for aBoc, we conducted tissue-specific rescue experiments of frpr-17 mutants. frpr-17 transcripts are detected in a subset of neurons, body wall muscle, and hmc[25,26]. frpr-17 expression in either neurons or muscles failed to rescue the aBoc defects of frpr-17 mutants (Fig. 1c), whereas frpr-17 expression in hmc under either the arg-1 promoter[28] or the nmur-3(Δ) promoter (Fig. S1a, b) fully restored wild-type aBoc frequencies to

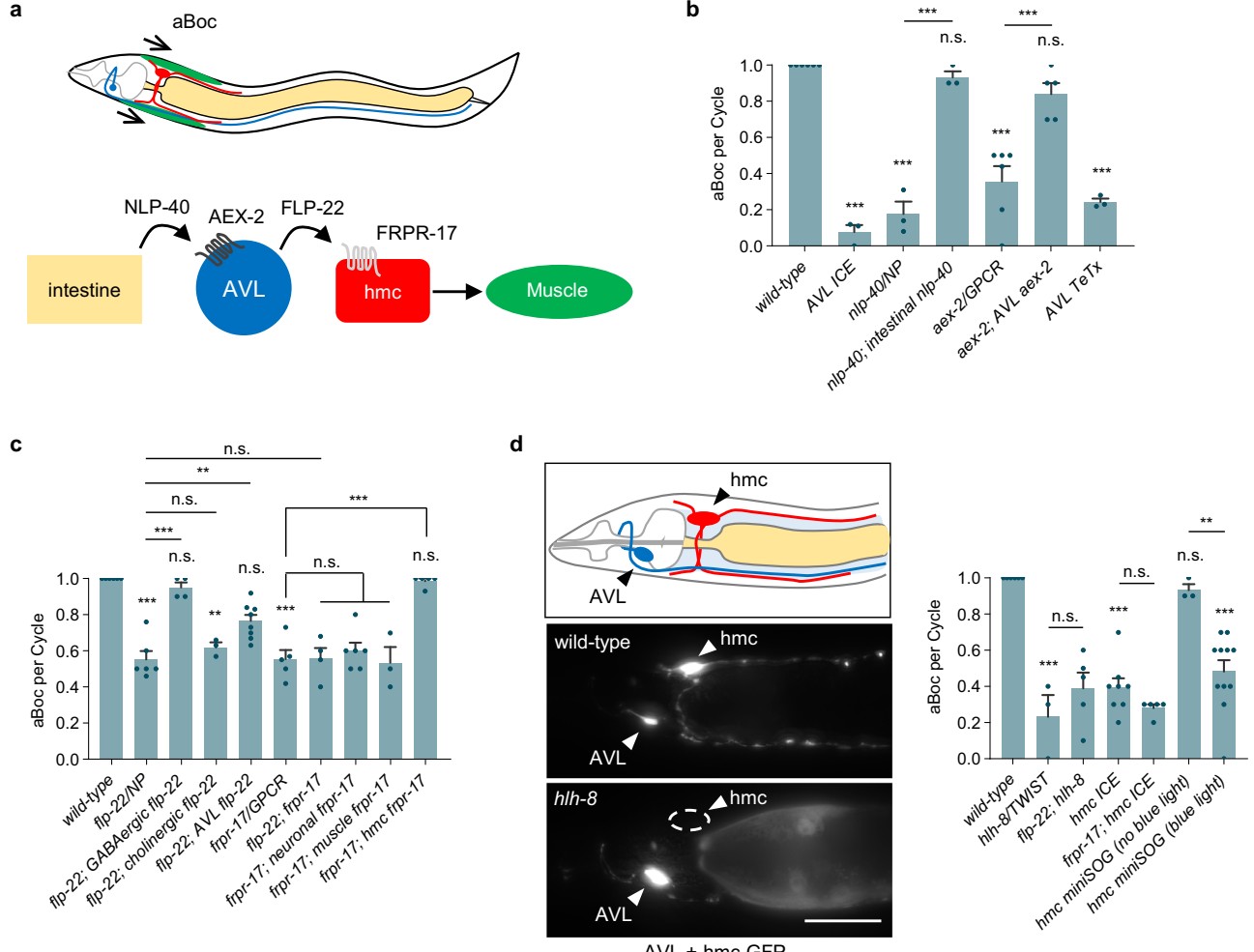

**Fig. 1 | AVL controls aBoc through peptidergic signaling to hmc. a** Schematic showing the circuit controlling aBoc identified in this study. **b** Quantification of the number of aBocs per defecation cycle in adult animals of the indicated genotypes. "*intestinal nlp-40*" denotes *nlp-40* cDNA expressed under the intestine-specific *ges-1* promoter. "*AVL ICE*", "*AVL aex-2*", and "*AVL TeTx*" denote ICE, *aex-2* cDNA or TeTx expressed in AVL using the *nmur-3(1 kb)* promoter. *n* = 6, 3, 3, 3, 6, 5, 3 independent animals. **c** Quantification of the number of aBocs per defecation cycle in adult animals of the indicated genotypes. *flp-22(vj229)* and *frpr-17(vj265)* null mutants were analyzed. "*GABAergic flp-22*" and "*cholinergic flp-22*" denote *flp-22* cDNA expressed under the GABAergic-specific *unc-47* and cholinergic-specific *unc-129* promoters, respectively. "*AVL flp-22*" denotes *flp-22* cDNA expressed in AVL using the *nmur-3(1 kb)* promoter. "*neuronal frpr-17*" denotes *frpr-17* cDNA expressed under the pan-neuronal *rab-3* promoter. "*muscle frpr-17*" denotes *frpr-17* cDNA expressed

under the muscle-specific *myo-3* promoter. "*hmc frpr-17*" denotes *frpr-17* cDNA expressed in hmc using the *nmur-3(Δ)* promoter. *n* = 6, 6, 4, 3, 8, 5, 4, 6, 3 independent animals. **d** *Left:* schematic of the location of AVL and hmc in the neck region and representative images showing GFP transcriptional reporters in wild-type and *hlh-8* mutants, scale bar, 40 μm. "AVL + hmc GFP" denotes expressing *GFP* under the *numr-3(3 kb)* promoter. GFP fluorescence is not detectable in *hlh-8* mutants (0 out of 20 animals exhibited GFP fluorescence in hmc). *Right:* quantification of the number of aBocs per defecation cycle in adult animals of the indicated genotypes. "*hmc ICE*" and "*hmc miniSOG*" denote transgenes expressing ICE or miniSOG in hmc under control of the *nmur-3(Δ)* promoter. *n* = 6, 3, 5, 8, 5, 3, 11 independent animals. Data are presented as mean values ± SEM. ***$P < 0.001$, **$P < 0.01$ and *$P < 0.05$ in one-way ANOVA with Bonferroni's correction for multiple comparisons; n.s. not significant.

*frpr-17* mutants (Fig. 1c and S2a). Thus, *frpr-17* functions in hmc to positively regulate aBoc.

## hmc controls aBoc

Several observations indicate that hmc is required for the execution of the aBoc step. First, *hlh-8* encodes the sole ortholog of the human basic helix-loop-helix domain containing mesodermal transcription factor TWIST, and *hlh-8* is required for hmc differentiation[27–29]. We found that in *hlh-8* null mutants, which are missing hmc (Fig. 1d), aBoc occurred in just ~35% of cycles (Fig. 1d). Second, genetic ablation of hmc by hmc-specific expression of ICE reduced aBoc frequency to a similar extent as in *hlh-8* mutations (Fig. 1d). Finally, we expressed mini singlet oxygen generator (miniSOG), a photosensitizer that generates reactive oxygen species after blue light illumination[39,40], selectively in hmc. miniSOG activation at the L4 stage (after hmc and AVL are

differentiated) led to aBoc defects one day later in adulthood similar to hmc-ablated animals (Fig. 1d). The aBoc frequency in hmc-ablated animals was not further reduced by *flp-22* or *frpr-17* mutations (Fig. 1d). Thus, hmc has a post-developmental role in promoting normal aBoc frequency, and *frpr-17* is likely to function exclusively in hmc to control aBoc.

## hmc is rhythmically activated in phase with AVL activation and aBoc

To determine whether hmc is rhythmically activated during the DMP, we constructed an imaging line in which the calcium indicator GCaMP was expressed simultaneously in the intestine, AVL, and hmc. Animals bearing this GCaMP array exhibited normal DMP timing, expulsion frequency, and aBoc frequencies (Fig. S2a, b, d), indicating that signaling of the aBoc circuit is likely to be normal in this strain. We

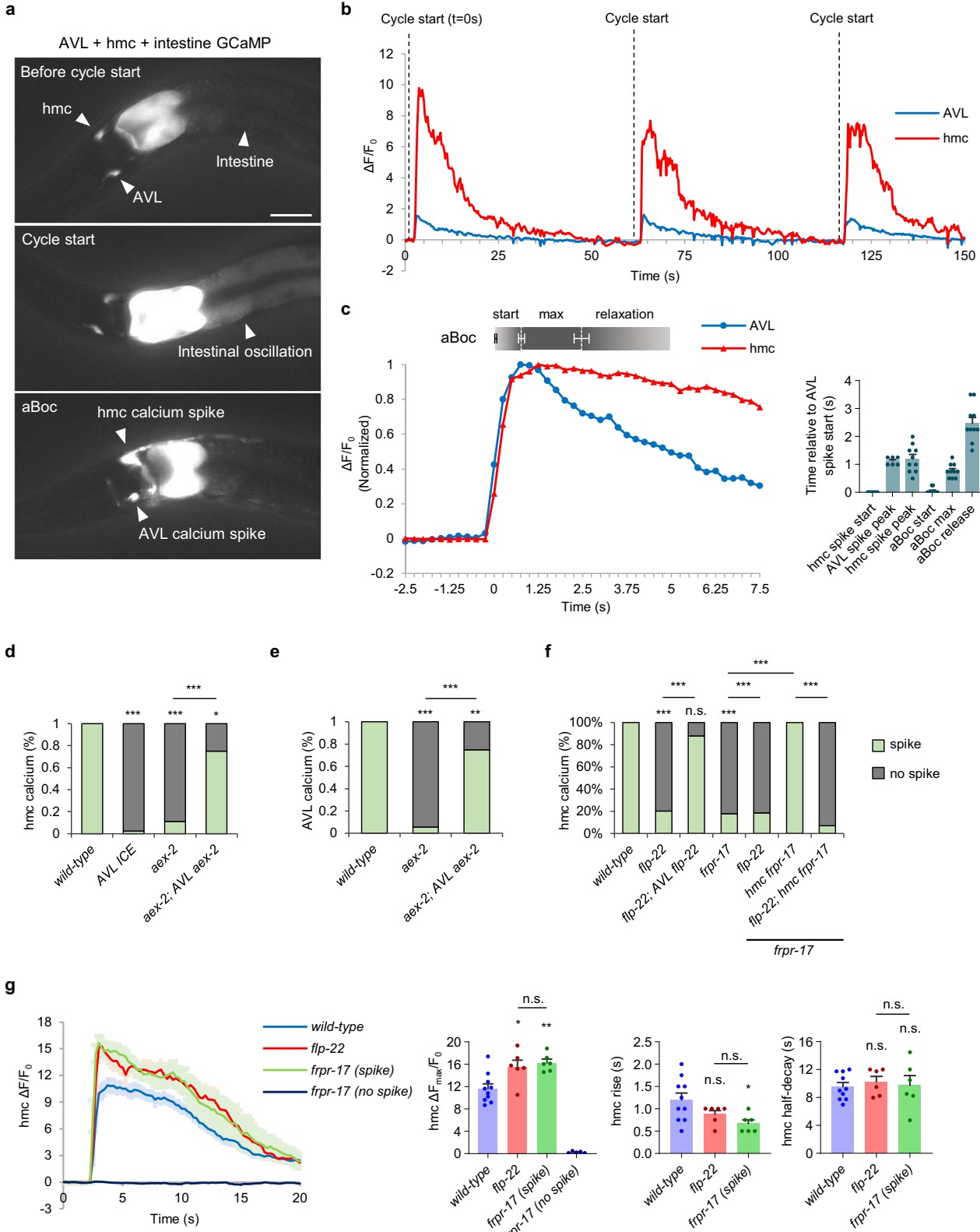

collected time-lapse images of the neck regions of freely moving animals and quantified fluorescence intensities in the intestine, the AVL soma, and the hmc cell body during the DMP (Fig. 2a and Supplementary Movie 1). In between cycles, baseline levels of GCaMP fluorescence in the intestine, AVL, and hmc were consistently low. About every 50 s, a calcium wave was observed in the intestine, marking the beginning of the cycle (Fig. 2a). About three seconds later, a GCaMP

fluorescent spike was observed in the AVL soma and its process, as well as in the hmc cell body and processes (Fig. 2a and b). The GCaMP fluorescence spikes were not an artifact of movement or muscle contraction since in animals co-expressing both GCaMP and mCherry, mCherry fluorescence in AVL and hmc remained at baseline levels throughout the cycle, including during aBoc (Fig. S3a). The AVL and hmc calcium spikes initiated within 250 ms of each other. AVL spikes

**Fig. 2 | hmc is rhythmically activated by FLP-22 and FRPR-17 signaling.**
**a** Representative images from videos showing GCaMP fluorescence in the intestine, AVL, and hmc before cycle start, at cycle start indicated by the intestinal calcium oscillation, and at the onset of AVL and hmc activation in animals expressing GCaMP3 in the intestine (under the *nlp-40* promoter) and GCaMP6 in AVL and hmc (under the *nmur-3(3 kb)* promoter). Scale bar, 40 µm. **b** Representative trace showing the calcium dynamics in the AVL soma and hmc cell body during three sequential cycles of the defecation motor program in adult animals. The calcium oscillation in the intestine marks the start of each cycle (dotted line). **c** *Left:* normalized GCaMP fluorescence intensities in AVL and hmc and aBoc contractions averaged from time lapse images from 5 cycles for AVL calcium spikes, 10 cycles for hmc calcium spikes, and 10 cycles for aBoc. The x axis denotes seconds relative to the initiation of the calcium spike in AVL. The times that the average aBoc contraction begins (start), reaches maximum contraction (max), and begins relaxation (relaxation) are indicated. *Right:* quantification of the indicated parameters relative to the onset of the calcium spike in AVL. Data are presented as mean values ± SEM. **d–f** Quantification of the number of calcium spikes per cycle observed in AVL and

hmc in adult animals of the indicated genotypes. "AVL *aex-2*" denotes *aex-2* cDNA expressed in AVL under the *nmur-3(1 kb)* promoter. "AVL *flp-22*" denotes *flp-22* cDNA expressed in AVL under the *nmur-3(1 kb)* promoter. "hmc *frpr-17*" denotes *frpr-17* cDNA expressed in hmc under the *nmur-3(Δ)* promoter. Wild-type: 33 cycles in 8 animals, *aex-2*: 36 cycles in 7 animals, *aex-2; AVL aex-2*: 20 cycles in 4 animals, AVL ICE: 43 cycles in 9 animals, *flp-22*: 49 cycles in 9 animals, *frpr-17*: 56 cycles in 13 animals, *flp-22; frpr-17*: 43 cycles in 9 animals, *frpr-17; hmc frpr-17*: 20 cycles in 7 animals, *flp-22; frpr-17; hmc frpr-17*: 42 cycles in 9 animals. ***P < 0.001 and *P < 0.05 in two-sided chi-square test with Bonferroni's correction for multiple comparisons; n.s., not significant. **g** *Left:* average traces of calcium dynamics in the indicated genotypes in hmc aligned to the calcium spike initiation time. The solid lines indicate average fold change in GCaMP intensity and the shades indicate SEM. *Right:* quantification of the average peak amplitude, rise time, and half-decay time. Data are presented as mean values ± SEM. *n* = 10, 6, 6, 5 cycles from different animals. **P < 0.01 and *P < 0.05 in one-way ANOVA with Bonferroni's correction for multiple comparisons; n.s. not significant.

---

peaked after about 1 s following spike initiation, averaging 2-fold increase from baseline, (Fig. 2,) and decayed with an average half-decay time of 4.63 s (Fig. 2). hmc spikes were significantly larger in peak amplitude (averaging 12-fold increase from baseline) and decayed more slowly than those in AVL before returning to baseline with an average half-decay time of 9.55 s (Fig. 2).

Each calcium spike was accompanied by an aBoc, which could be seen in the time-lapse images as a rapid posterior-directed displacement of the pharynx into the anterior intestine (Fig. S3b). On average, aBocs initiated slightly after the initiation of the AVL and hmc calcium spikes and reached maximal contraction slightly before the peak of both spikes (Fig. 2c). Maximal contractions lasted for on average 1.7 s followed by a slower more variable relaxation lasting for a few seconds (Fig. 2c), which is similar to the timing of the contraction-relaxation of aBocs as observed under the dissecting microscope[20]. Thus, hmc is rhythmically activated in phase with the calcium spikes in the intestine and AVL, and the timing of aBoc correlates with that of hmc activation. Together, these results indicate that AVL and hmc function together in a rhythmically activated circuit to control aBoc.

## AVL activates hmc, which in turn activates neck muscles

Our live calcium imaging analysis showed that AVL and hmc initiated activation and reached peak amplitude at about the same time (Fig. 2c). To determine if AVL activates hmc or vice versa, we examined calcium responses in animals in which activation of AVL was disrupted. We reasoned that if AVL activates hmc, disrupting AVL activation should abolish calcium responses in hmc, however if hmc activates AVL, disrupting AVL activation should have no effect on calcium dynamics in hmc. To test this idea, we first genetically ablated AVL by expressing ICE selectively in AVL. We found that AVL ablation reduced hmc calcium spike frequency from 100% to less than 10% per cycle (Fig. 2d). Similarly, silencing AVL (in *aex-2* mutants), reduced the frequency of calcium spikes in AVL from 100% to 10% of cycles, and led to a corresponding reduction in hmc activation frequency (Fig. 2d, e). The defects in activation of both AVL and hmc in *aex-2* mutants could be rescued by expressing *aex-2* cDNA selectively in AVL (Fig. 2d and e). Thus, hmc cannot be activated when AVL is silenced, placing AVL upstream of hmc in this circuit.

To test whether hmc activation controls neck muscle contraction or vice versa, we examined mutants in *unc-54*, which encodes myosin heavy chain and is required for calcium-dependent contraction of body wall muscles[41]. As expected, *unc-54* loss-of-function mutants had very weak aBocs, yet we always observed calcium responses in hmc that were similar in frequency and amplitude to those of wild-type controls (Fig. S3c, d, Supplementary Movie 4). Thus, hmc can be activated even when neck muscle contraction is severely compromised, indicating that the neck muscles are unlikely to activate hmc.

Together, we conclude that AVL is upstream of hmc and hmc is upstream of neck muscle in the aBoc circuit (Fig. 1a).

## FLP-22 released from AVL can bind to hmc in vivo

Serial electron microscopy shows that the proximal axon of AVL lies in close proximity (within 5 um) of the ventral hmc process in the neck region, and that these processes are separated by the pseudocoelom[22]. Since neuropeptides can be secreted into the pseudocoelomic fluid[19,23], we wanted to test whether FLP-22 is secreted into the pseudocoelom and whether it has access to hmc. First, we expressed FLP-22 fusion proteins with the pH sensitive GFP variant pHluorin (FLP-22::pHluorin) in DCVs in AVL. pHluorin is quenched in DCV lumens due to their acidity, but pHluorin is unquenched in mutants lacking *unc-32*, which encodes the $V_0$ subunit of the vesicular ATPase proton transporter responsible for acidifying DCVs[42]. Fluorescence was not visible in AVL axons of wild-type animals expressing FLP-22::pHluorin, but a highly punctate pattern of fluorescence characteristic of DCVs was visible in the proximal AVL axon of *unc-32* mutants (Fig. S4a), indicating that FLP-22 is packaged into DCVs in the proximal AVL axon.

Next, to determine whether secreted FLP-22 has access to hmc, we crossed the FLP-22::pHluorin transgene into animals expressing a GFP-binding domain (GBD) specifically on the surface of hmc (GBD::SAX-7[43]). In animals co-expressing both FLP-22::pHluorin in AVL and GBD::SAX-7 in hmc, strong fluorescence was seen along the entire surface of hmc, including the dorsal and ventral hmc processes as well as the cell body (Fig. S4b). Control animals expressing GBD::SAX-7 alone exhibited no fluorescence (Fig. S4c). Thus, FLP-22 is likely to be released from DCVs in AVL into the pseudocoelomic cavity, where it can diffuse to hmc.

## FLP-22 released from AVL activates *frpr-17* in hmc

To determine whether *flp-22* signaling activates hmc, we conducted live calcium imaging of *flp-22* mutants. Calcium spike timing, frequency, and dynamics in AVL were normal in *flp-22* mutants (Fig. S5a–c), but calcium spikes in hmc occurred in just 20% of cycles (Fig. 2f). In these cycles, calcium spike onset was slightly delayed (Fig. S5d) and larger in amplitude compared to wild-type controls (Fig. 2g). The reduced hmc calcium spike frequency of *flp-22* mutants could be nearly fully rescued by expressing *flp-22* cDNA selectively in AVL (Fig. 2f). Thus, AVL is likely to be an endogenous source of *flp-22* for hmc activation.

*frpr-17* mutants exhibited similar calcium phenotypes as *flp-22* mutants: calcium spike timing, frequency, peak amplitude and rise time in AVL were normal (Fig. S5a–c), but calcium spikes in hmc occurred in just 20% of cycles (Fig. 2f and Supplementary Movie 2), and in these cycles, calcium spike onset was slightly delayed (Fig. S5d) and larger in amplitude compared to wild-type controls (Fig. 2g). The

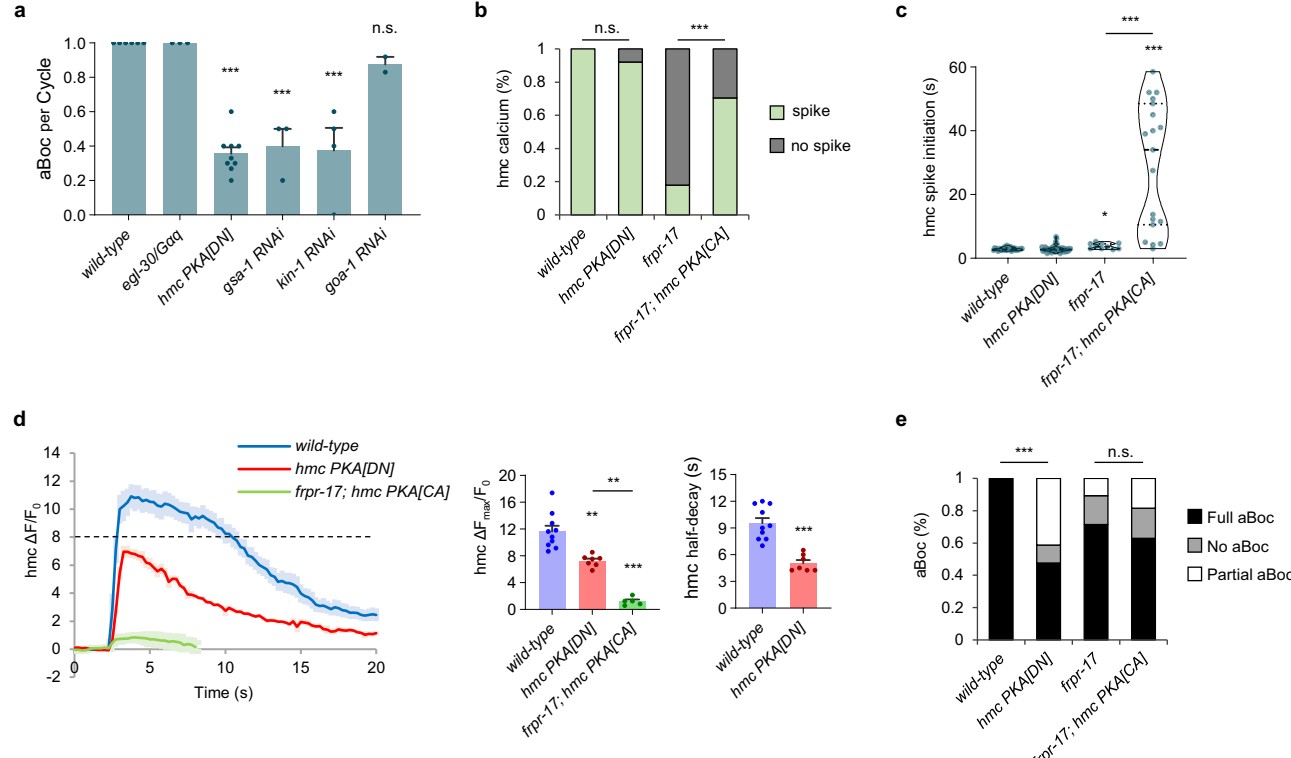

**Fig. 3 | A PKA signaling cascade functions in hmc to facilitate hmc activation.**
**a** Quantification of the number of aBocs per cycle in adult animals of the indicated genotypes. "*hmc PKA[DN]*" denotes a transgene in which a *PKA* dominant negative variant is expressed in hmc using the *nmur-3(Δ)* promoter. RNAi denotes RNA interference-mediated knockdown of the indicated gene by feeding. Data are presented as mean values ± SEM. *n* = 6, 3, 9, 3, 4, 2 independent animals. ***P < 0.001 in one-way ANOVA with Dunnett's correction for multiple comparisons; n.s. not significant. **b** Quantification of the number of hmc calcium spikes observed per cycle in adult animals of the indicated genotype. "*hmc PKA[DN]*" and "*hmc PKA[CA]*" denote transgenes expressing *PKA* dominant negative or *PKA* constitutively active variants in hmc under the control of the *nmur-3(Δ)* promoter. For (**b**) and (**e**), wild-type: 33 cycles in 8 animals, *hmc PKA[DN]*: 63 cycles in 21 animals, *frpr-17*: 56 cycles in 13 animals, *frpr-17; hmc PKA[CA]*: 27 cycles in 6 animals. ***P < 0.001 in two-sided chi-square test with Bonferroni's correction for multiple comparisons; n.s. not significant. **c** Violin plots of calcium spike initiation time in hmc after the end of the intestinal calcium oscillation. Dashed lines refer to median and dotted lines refer to quartiles. ***P < 0.001 and *P < 0.05 in Kruskal–Wallis test with Dunn's correction for multiple comparisons. **d** *Left:* average traces of calcium dynamics in hmc aligned to the calcium spike initiation time in the indicated mutants. "*hmc PKA[DN]*" and "*hmc PKA[CA]*" denote transgenes expressing *PKA* dominant negative or *PKA* constitutively active variants in hmc under the control of the *nmur-3(Δ)* promoter. The solid lines indicate average fold change in GCaMP intensity and the shades indicate SEM. Dotted line represents a possible critical threshold for aBoc. *Right:* quantification of the average peak amplitude and half-decay time. Data are presented as mean values ± SEM *n* = 10, 7, 5 cycles from different animals. ***P < 0.001 and **P < 0.01 in one-way ANOVA with Bonferroni's correction for multiple comparisons and in two-tailed Student's *t*-test. **e** Quantification of the number aBocs observed per cycle in the indicated mutants in the time lapse calcium images. "*hmc PKA[DN]*" and "*hmc PKA[CA]*" denote transgenes expressing *PKA* dominant negative or *PKA* constitutively active variants in hmc under the control of the *nmur-3(Δ)* promoter. ***P < 0.001 in two-sided chi-square test with Bonferroni's correction for multiple comparisons; n.s. not significant.

reduced calcium spike frequency of *frpr-17* mutants could be fully rescued by expressing *frpr-17* selectively in hmc, and this rescue was abolished by *flp-22* mutations (Fig. 2f). *flp-22; frpr-17* double mutants exhibited similar calcium defects as *flp-22* or *frpr-17* single mutants, with calcium spike frequency in hmc of 20% (Fig. 2f). Taken together, these results reinforce the notion that hmc functions downstream of AVL, since AVL can activate normally when hmc is selectively silenced. We conclude that activation of FRPR-17 in hmc by FLP-22 from AVL provides the predominant drive for hmc activation.

Examination of aBocs in the time lapse images of *flp-22* or *frpr-17* mutants revealed that in the 20% of cycles in which hmc was activated, aBocs always occurred, and these aBocs were indistinguishable from wild-type (termed "full aBocs", Fig. S3b). In cycles in which hmc was not activated, aBoc frequency was severely reduced, with full aBocs occurring in just half the cycles (Fig. S3b). Closer examination of time lapse images revealed that in many cycles in which hmc was not activated, aBocs that were smaller or lasted a shorter duration than wild-type aBocs (termed "partial aBocs") could be seen (Fig. S3b). The presence of full and partial aBocs in cycles without hmc activation reveals the existence of a mechanism that can promote aBoc independently of hmc activation (see below).

**Protein kinase A functions in hmc to control hmc activation**
GPCR signaling cascades commence with the activation of GPCR-coupled heterotrimeric G proteins. To identify the signaling cascade downstream of *frpr-17*, we first examined three Gα subunits of the heterotrimeric G proteins, *gsa-1*/Gαs, *goa-1*/Gαi/o, and *egl-30*/Gαq that are the most highly expressed in hmc[25]. RNAi-mediated knockdown of *gsa-1*/Gαs caused significant reductions in aBoc frequency, whereas knockdown of *goa-1*/Gαi/o or loss-of-function mutations in *egl-30*/Gαq caused no defects (Fig. 3a). Gαs can activate an evolutionarily conserved adenylyl cyclase – protein kinase A (PKA) signaling cascade[44]. RNAi-mediated knockdown of the PKA catalytic subunit, *kin-1*, reduced aBoc frequency to 38% (Fig. 3a), and hmc-specific expression of dominant negative PKA[DN] transgenes[45,46], reduced aBoc frequency to about 35% (Fig. 3a), which is similar to the reduction in aBoc frequency following hmc ablation (Fig. 1d). Together, these results suggest that FRPR-17 may control aBoc by activating PKA in hmc.

To determine how PKA signaling controls aBoc, we examined calcium dynamics in animals expressing PKA[DN] transgenes in hmc. Calcium spike frequency and initiation time in hmc were similar in PKA[DN] transgenic animals and non-transgenic controls (Fig. 3b and c), but the average calcium spike amplitude was reduced

by 40%, and the average time to half-decay was significantly faster than controls (Fig. 3d). In PKA[DN] animals, hmc calcium spikes were accompanied by a full aBoc in only about half of the cycles (Fig. 3e). These results suggest that PKA signaling in hmc controls aBoc by increasing calcium spike amplitude and/or duration above a certain threshold needed for full aBoc.

To determine whether PKA activity in hmc is sufficient to activate hmc, we expressed constitutively active (PKA[CA]) transgenes[46,47] in hmc. PKA[CA] transgenes increased the calcium spike frequency of *frpr-17* mutants from 20% to 70% (Fig. 3b). The calcium spikes did not occur at the normal time, but instead were "ectopic", occurring at random times during cycle intervals (Fig. 3c). The average amplitude of the ectopic calcium spikes was 20% of that of wild-type calcium spikes, and these spikes were never associated with aBoc (Fig. 3d and e). Thus, unregulated PKA activation can lead to sub-threshold calcium spike generation in hmc at random times. Taken together, these results suggest that the activation of hmc is not an all-or-none response, and that the PKA signaling in hmc may ensure that calcium spikes reach a threshold needed to trigger aBoc.

### The gap junction protein UNC-9 functionally couples hmc and neck muscles to promote aBoc

Prior ultrastructural studies show that the hmc cell body and processes form extensive gap junctions with neck muscles (Fig. 4a and ref. 22). Gap junctions are composed of multimers of subunits (termed connexins in vertebrates and innexins in invertebrates) that form hemichannels, which when coupled with hemichannels from neighboring cells form gap junctions. *C. elegans* encodes 25 innexins, four of which, *unc-9*, *inx-7*, *inx-10*, and *inx-11* are expressed at high levels in hmc[25]. Null mutations in *unc-9*/innexin, which is the most highly expressed innexin in hmc, caused significant reductions in aBoc frequency (32%, Fig. 4b), while null mutations in the other three innexins did not cause aBoc defects (Fig. S6a). Expression of *unc-9* cDNA selectively in hmc rescued the aBoc defects of *unc-9* mutants, whereas expression of *unc-9* cDNA in muscles failed to rescue (Fig. 4b). Since the aBoc frequency of *unc-9* mutants is similar to that of hmc ablated animals, aBoc arising from hmc activation is likely to occur exclusively through *unc-9*. Together, these results indicate that UNC-9/innexin is a component of the hemichannel on the hmc side of the gap junctions that is critical for aBoc.

Functional UNC-9::mTur2 fusion proteins adopted a highly punctate pattern of fluorescence in both dorsal and ventral processes of hmc as well as in the hmc cell body (Fig. 4c). In contrast, FRPR-17::Venus fusion proteins adopted a diffuse pattern of localization throughout the dorsal and ventral processes and surrounding the hmc cell body (Fig. 4c). These patterns are consistent with UNC-9 localizing to gap junctions and FRPR-17 residing on the plasma membrane.

If gap junctions couple hmc activation to neck muscle contraction by allowing calcium to pass from hmc into neck muscles, we predict that *unc-9* mutations should disrupt aBoc without altering hmc activation. We found that calcium spike frequencies in both AVL and hmc were normal in *unc-9* mutants, and calcium spike amplitudes and decay times in hmc were indistinguishable from wild-type controls (Fig. 4d, e, and S6b, Supplementary Movie 3). However, *unc-9* mutants exhibited dramatic reductions in aBoc frequency: full aBocs were observed in just 40% of cycles, and aBocs were absent or partial in the remaining cycles (Fig. 4f). Thus, *unc-9* does not contribute to hmc activation, but instead is critical for activated hmc to elicit aBoc. These results suggest that UNC-9-containing gap junctions couple hmc activation with neck muscle contraction by mediating the passage of calcium (and/or other signals) from hmc into neck muscles.

### Examining the roles of additional signaling pathways in hmc activation

To identify additional GPCR signaling pathways that might regulate hmc activity during aBoc, we examined mutants corresponding to a number of highly expressed putative GPCRs in hmc[25]. Null mutants in five of these genes, the neuropeptide GPCRs *nmur-3*, *npr-23*, *T11F9.1*, and *frpr-4*, as well as the orphan class D GPCR *srd-32* exhibited calcium spike frequencies in hmc that were similar to wild-type controls, and they did not significantly alter the hmc calcium spike frequency defects caused by *frpr-17* mutations (Fig. S7a, b). Mutations in these genes did not significantly alter rise time, peak amplitude, or half decay time of hmc calcium spikes, with the exception of *npr-23* mutants, which increased the half decay time of hmc calcium spikes, and *nmur-3* mutants, which decreased peak spike amplitude (Fig. S7c). Similarly, we found that *unc-25*/glutamate decarboxylase mutants exhibited similar calcium spike frequencies in AVL and hmc as wild-type controls and did not alter hmc activation or aBoc frequency of *frpr-17* mutants (Fig. S7d). Thus, *nmur-3*, *npr-23*, *T11F9.1*, *frpr-4*, *srd-32* and GABA are likely to play no or minor roles in hmc activation either in the presence or absence of *frp-17* signaling.

### hmc activity is negatively regulated by FLP-9/FMRFamide and FRPR-21/GPCR signaling in hmc

*frpr-21* encodes the second most highly expressed neuropeptide GPCR in hmc after *frpr-17* (Fig. S7a). *frpr-21* putative null mutants had normal egg laying and locomotion rates (Fig. S2e, f), and they exhibited aBoc frequencies and calcium spike frequencies in both AVL and hmc that were similar to wild-type controls (Fig. 5a, b and S8a). *frpr-21* mutations did not significantly alter rise time, peak amplitude, or half decay time of calcium spikes in either AVL or hmc (Fig. 5c and S8b). However, *frpr-21* mutations significantly increased the calcium spike frequency in *frpr-17* mutants from 20% to 50% (Fig. 5b). The restored calcium spikes occurred in phase with AVL activation, but like the calcium spikes in *frpr-17* mutants, their onset was slightly delayed (Fig. S8c). Expression of *frpr-21* cDNA in hmc fully reverted the calcium spike frequency of *frpr-21; frpr-17* double mutants (Fig. 5b), suggesting that *frpr-21* functions in hmc to inhibit hmc activation. FRPR-21::Venus fusion proteins adopted a diffuse pattern of fluorescence in the hmc cell body and processes, consistent with surface expression in hmc (Fig. 5d). *flp-22; frpr-21* double mutants had slightly higher aBoc frequency compared to *flp-22* mutants, but this increase did not reach significance (Fig. 5a).

The FRPR-21 GPCR has no reported ligand. *flp-9* encodes an FMRF-like peptide that is predicted to be processed into two identical peptides that have been isolated from *C. elegans* lysates[38]. Synthetic FLP-9 peptides relax muscles and reduce cAMP levels when injected into the nematode *Ascaris suum*[48]. We found that like *frpr-21* mutations, *flp-9* null mutations did not cause defects in egg laying or locomotion (Fig. S2e, f). *flp-9* mutations did not alter aBoc frequencies or calcium dynamics in hmc on their own (Fig. 5e, f), but they restored calcium spike frequency to *frpr-17* mutants from 20% to 50% (Fig. 5b). Expression of *flp-9* cDNA under an endogenous promoter fragment (P*flp-9::flp-9*) fully reverted the hmc activation phenotype of *flp-9; frpr-17* double mutants to 20% (Fig. 5b). Similarly, expression of *flp-9* cDNA selectively in GABAergic neurons (P*unc-47::flp-9*) fully reverted the hmc activation phenotype of *flp-9; frpr-17* double mutants to 20% (Fig. 5b) Thus, FLP-9 secretion from the GABAergic nervous system is sufficient to negatively regulate hmc activation in the absence of *frpr-17* signaling.

FLP-9 is reported to interact with four neuropeptide GPCRs in in vitro binding assays (DMSR-1, DMSR-7, EGL-6 and FRPR-8), and FLP-22 is reported to interact with DMSR-7[49]. Putative null mutants in each of these GPCRs did not alter hmc activation frequency in wild-type or in *frpr-17* mutant backgrounds (Fig. S8d), suggesting that FLP-9 and FLP-22 are unlikely to control hmc activation via these GPCRs. Instead, FLP-9 is likely to be the ligand for FRPR-21 in regulating aBoc. First, *flp-9; frpr-21; frpr-17* triple mutants had a similar hmc calcium spike phenotype as *flp-9; frpr-17* or *frpr-21; frpr-17* double mutants (Fig. 5b). Second, overexpressing *frpr-21* cDNA selectively in hmc (*frpr-21 (OE)*) or overexpressing *flp-9* cDNA in the GABAergic neurons (*flp-9 (OE)*) resulted in

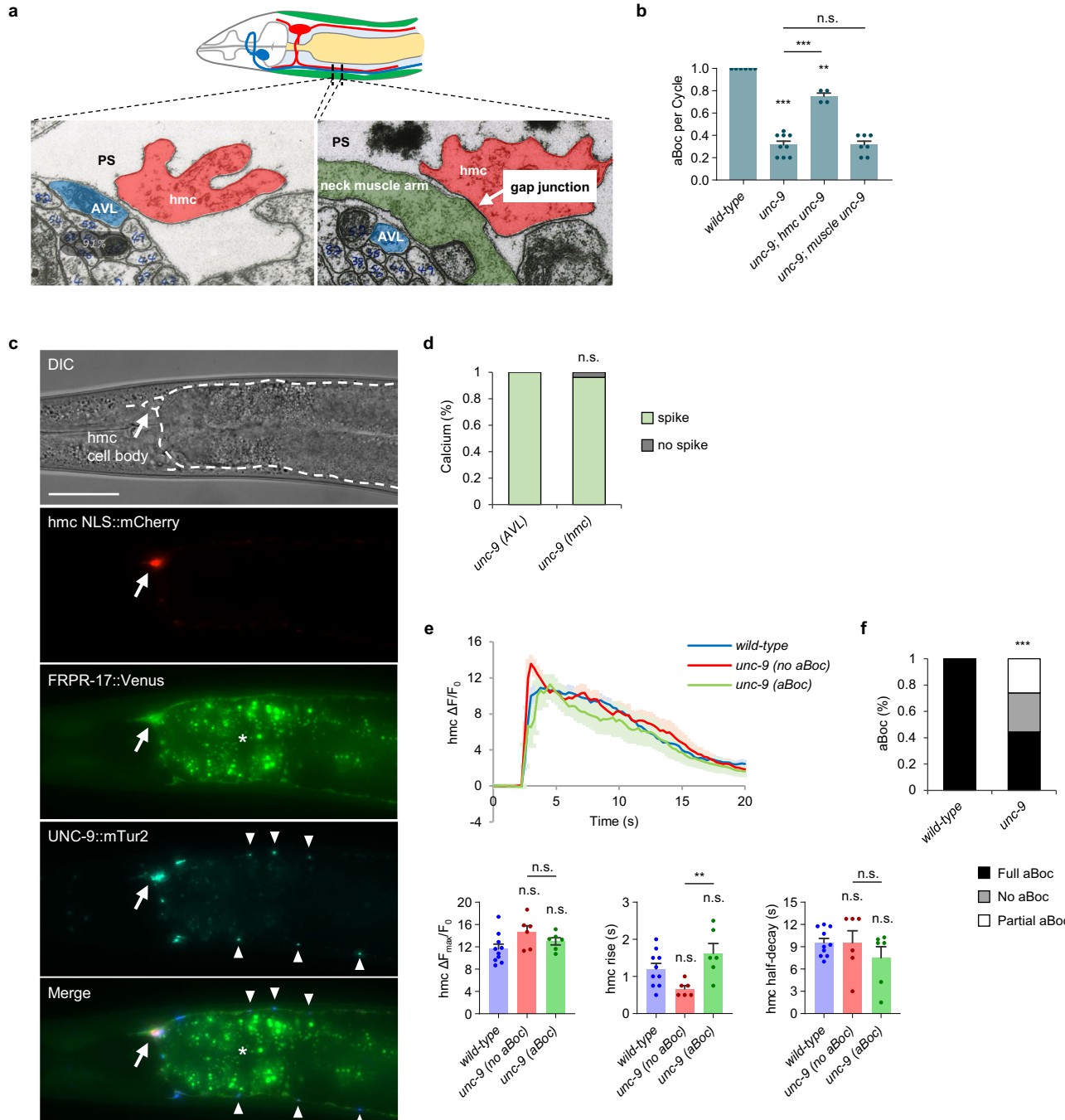

**Fig. 4 | Gap junctions composed of UNC-9/innexin are required for hmc signaling to neck muscles. a** Transmission electron micrograph images of cross sections on the ventral side of the neck showing the AVL and hmc processes, adapted from[22]. AVL and hmc are separated by the pseudocoelom (PS). A large gap junction appears as an electron dense area where the plasma membranes of the neck muscle arm and hmc contact each other. **b** Quantification of the number of aBocs per cycle in adult animals of the indicated genotypes. "*hmc unc-9*" and "*muscle unc-9*" denote expressing *unc-9::mTur2* fusion proteins or *unc-9* cDNA under the *nmur-3(Δ)* and *myo-3* promoter, respectively. Data are presented as mean values ± SEM. *n* = 6, 9, 4, 7 independent animals. \*\*\**P* < 0.001 and \*\**P* < 0.01 in one-way ANOVA with Bonferroni's correction for multiple comparisons; n.s. not significant. **c** Representative images of the neck region of an animal co-expressing NLS::mCherry (arrow), FRPR-17::Venus (plasma membrane of hmc), and UNC-9::mTur2 (arrowheads) in hmc under the *nmur-3(Δ)* promoter. The dotted line in the top panel shows the position of hmc. The asterisk denotes autofluorescence in

the intestine. Similar expression patterns were observed in all 16 animals examined. Scale bar, 40 μm. **d** Quantification of the number of calcium spikes observed in AVL and hmc during DMP in adult *unc-9* mutants. For **d** and **f**, *unc-9*: 27 cycles in 9 animals. Two-sided Fisher's exact test; n.s., not significant. **e** Average traces of calcium dynamics in hmc aligned to the calcium spike initiation time in the indicated mutants. The solid lines indicate average fold change in GCaMP intensity and the shades indicate SEM. Quantification of the average peak amplitude, rise time, and half-decay time. *unc-9 (no aBoc)* refers to cycles with calcium spike followed by no aBoc and *unc-9 (aBoc)* refers to cycles with calcium spike followed by aBoc. Data are presented as mean values ± SEM. *n* = 10, 6, 6 cycles from different animals. \*\**P* < 0.01 in one-way ANOVA with Bonferroni's correction for multiple comparisons; n.s. not significant. **f** Quantification of the number aBocs observed per cycle in the indicated mutants in the time lapse calcium images. \*\*\**P* < 0.001 in two-sided chi-square test.

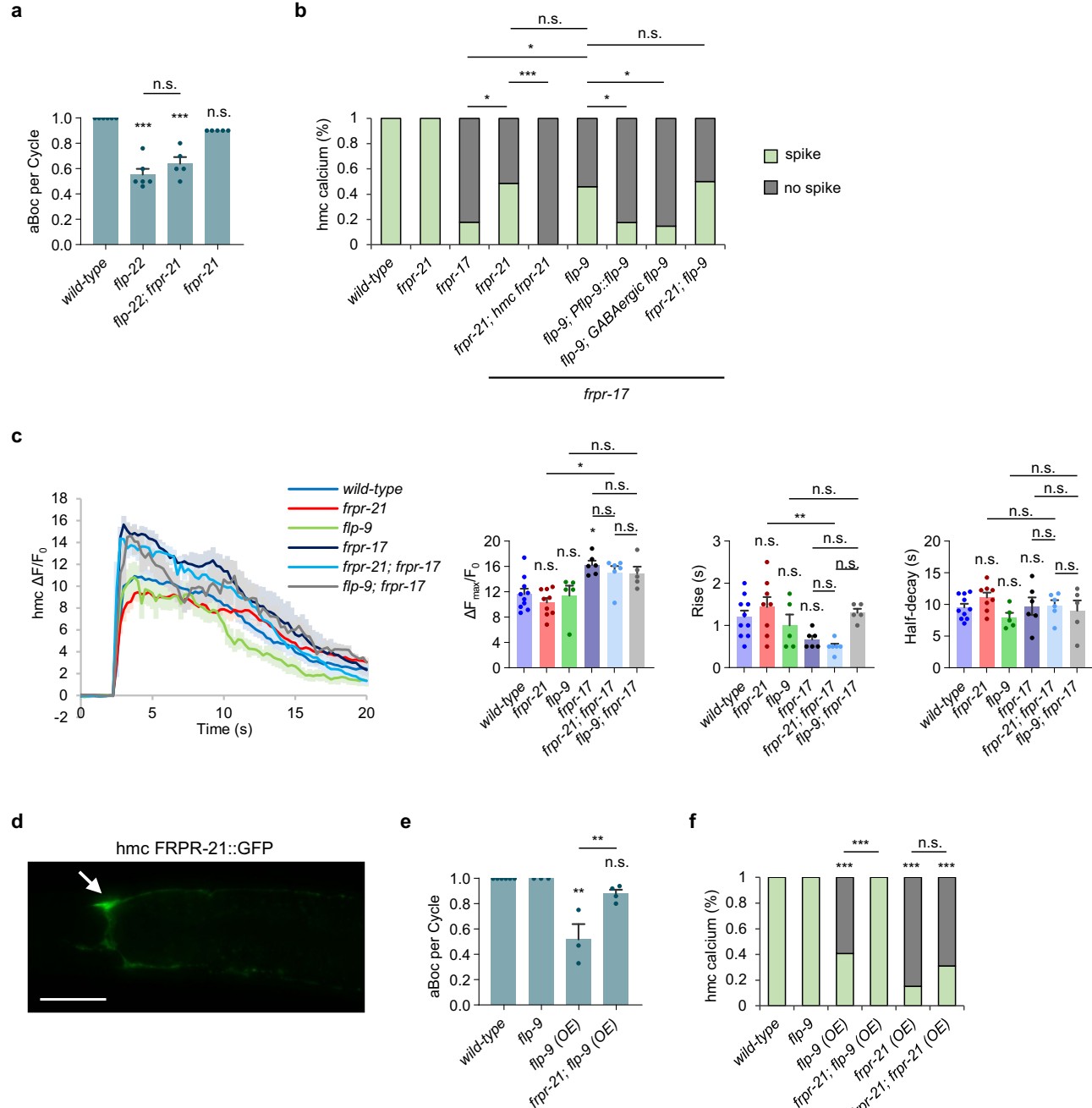

**Fig. 5 | hmc activity is negatively regulated by the *flp-9* FMRFamide-like neuropeptide and the *frpr-21* GPCR. a** Quantification of the number of aBocs per cycle in adult animals of the indicated genotypes. Data are presented as mean values ± SEM. *n* = 6, 6, 5, 5 independent animals. ***P* < 0.001 in one-way ANOVA with Bonferroni's correction for multiple comparisons; n.s. not significant. **b** Quantification of the number of calcium spikes observed per cycle in hmc in adult animals of the indicated genotypes. "hmc *frpr-21*" denotes expressing *frpr-21* cDNA under the *nmur-3(Δ)* promoter. "P*flp-9 flp-9*" denotes expressing *flp-9* cDNA under a 4.8 kb *flp-9* promoter fragment. "GABAergic *flp-9*" denotes expressing *flp-9* cDNA under the *unc-47* promoter. Wild-type: 33 cycles in 8 animals, *frpr-21*: 23 cycles in 9 animals, *frpr-17*: 56 cycles in 13 animals, *frpr-21; frpr-17*: 32 cycles in 10 animals, *frpr-21; frpr-17; hmc frpr-21*: 38 cycles in 7 animals, *flp-9; frpr-17*: 47 cycles in 8 animals, *frpr-21; flp-9; frpr-17*: 44 cycles in 7 animals. ***P* < 0.001 and **P* < 0.05 in two-sided chi-square test with Bonferroni's correction for multiple comparisons; n.s. not significant. **c** *Left*: average traces of calcium dynamics in hmc aligned to the calcium spike initiation time in the indicated mutants. The solid lines indicate average fold change in GCaMP intensity and the shades indicate SEM. *Right*: quantification of the average peak amplitude, rise time, and half-decay time from the traces on the left. Data are

presented as mean values ± SEM. *n* = 10, 9, 5, 6, 6, 5 cycles from different animals. ***P* < 0.01 and **P* < 0.05 in one-way ANOVA with Bonferroni's correction for multiple comparisons; n.s. not significant. **d** Representative image of hmc cell body (arrow) and processes from adults expressing FRPR-21::GFP in hmc under the *nmur-3(Δ)* promoter. Similar expression patterns were observed in all 15 animals examined. Scale bar, 40 µm. **e** Quantification of the number of aBocs per cycle in adult animals of the indicated genotypes. "*flp-9 (OE)*" denotes expression of *flp-9* cDNA under the GABAergic-specific *unc-47* promoter. Data are presented as mean values ± SEM. *n* = 6, 3, 3, 4 independent animals. ***P* < 0.01 in one-way ANOVA with Bonferroni's correction for multiple comparisons; n.s. not significant. **f** Quantification of the number of calcium spikes observed per cycle in hmc in adult animals of the indicated genotypes. "*frpr-21 (OE)*" denotes expressing *frpr-21* cDNA in hmc using the *nmur-3(Δ)* promoter. Wild-type: 33 cycles in 8 animals, *flp-9*: 27 cycles in 8 animals, *flp-9 (OE)*: 27 cycles in 5 animals, *frpr-21; flp-9 (OE)*: 23 cycles in 5 animals, *frpr-21 (OE)*: 46 cycles in 7 animals, *frpr-21; frpr-21 (OE)*: 29 cycles in 8 animals. ***P* < 0.001 in two-sided chi-square test with Bonferroni's correction for multiple comparisons; n.s. not significant.

missing hmc calcium spikes in at least 60% of cycles (Fig. 5f). Finally, *flp-9 (OE); frpr-21* double mutants had wild-type aBoc and hmc calcium spike frequency (Fig. 5e, f), indicating that the defects caused by FLP-9 overexpression are mediated by *frpr-21*. Thus, *flp-9* and *frpr-21* function in a common genetic pathway, and *frpr-21* functions downstream of *flp-9* to inhibit aBoc by negatively regulating hmc activation. Because *frpr-21* or *flp-9* mutations restore calcium transients at the proper time in the cycle (albeit slightly delayed) in the absence of *frpr-17* (Fig. S8c), we conclude that *frpr-21* inhibits rhythmic hmc activation by a mechanism that is independent of *frpr-17* signaling.

## Discussion

Here we identified a simple circuit activated by peptidergic signaling that links pacemaker activity to muscle contraction during a rhythmic behavior. This circuit is composed of four tissues: the intestine, AVL, hmc, and neck muscles. Based on our findings, we suggest the following model for how this circuit controls aBoc (Fig. 6). The calcium wave in the intestine every 50 s leads to the secretion of the pacemaker signal NLP-40 and the subsequent activation of the AEX-2/GPCR on AVL leading to the generation of a calcium transient in AVL. AVL

activation leads to the calcium-dependent secretion of the FMRFamide-like neuropeptide FLP-22 from the AVL axon, and FLP-22 subsequently binds to the FRPR-17/GPCR on hmc. Once activated, FRPR-17 functions upstream of a *gsa-1*/Gαs, *acy-1*/adenylate cyclase, *kin-1*/PKA signaling pathway in hmc that results in the generation of a calcium transient in hmc. Gap junctions composed of UNC-9/innexin in hmc couple the calcium transient in hmc with neck muscle contraction. Signaling by the *frpr-21* GPCR, activated by the FMRFamide-like neuropeptide *flp-9*, negatively regulates the generation of calcium transients in hmc. *flp-9* signaling may inhibit a yet unidentified activating pathway in hmc that functions in parallel with *frpr-17*, or *flp-9* signaling may negatively regulate the excitability of hmc by a less direct mechanism.

AVL has a well-characterized function as a motor neuron that controls the Exp step of the DMP through the release of GABA from NMJs[30,31]. Here, we have uncovered a second function for AVL in controlling aBoc through the release of FLP-22. Thus, AVL is a bifunctional neuron in which a single input (a calcium transient triggered by AEX-2/GPCR signaling) leads to divergent output signals to regulate two independent (but coordinated) behaviors through the release of

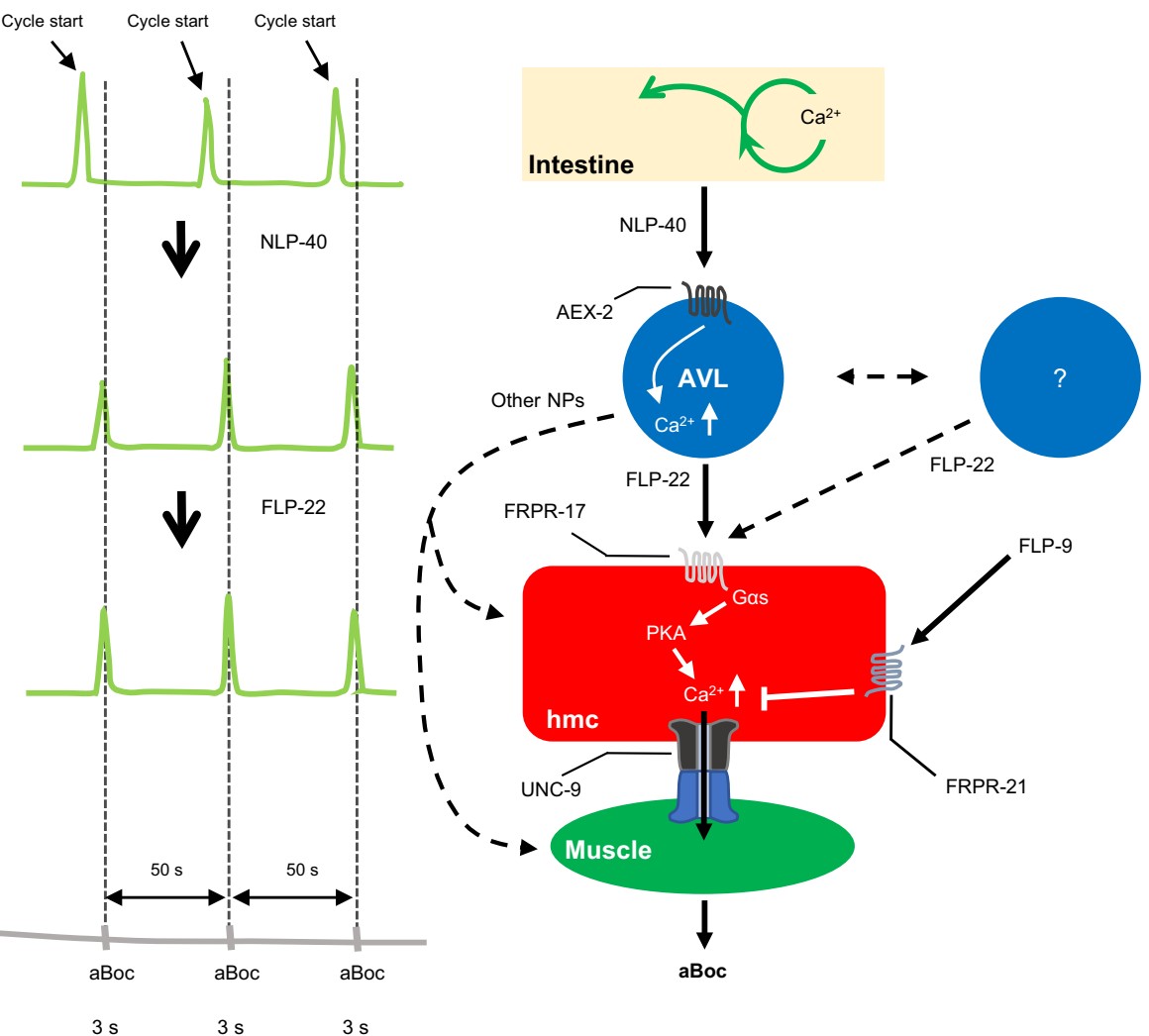

**Fig. 6 | Working model for the aBoc step.** During the defecation motor program, a calcium oscillation occurs in the intestine every 50 s, which leads to NLP-40 release from the intestine. NLP-40 activates its receptor, AEX-2, on AVL which results in a calcium spike. The calcium spike triggers secretion of the neuropeptide FLP-22 from AVL, which in turn activates its receptor, FRPR-17, on hmc. Activation of FRPR-17 leads to a calcium spike in hmc possibly by activating a signaling cascade composed of, GSA-1/Gαs and KIN-1/PKA. The calcium spike in hmc is transmitted to muscles through the gap junction protein UNC-9, resulting in a contraction of anterior body-wall muscles which is aBoc. Additional GABAergic neuron(s) may also secrete FLP-22 to activate hmc. FLP-9 activates its receptor, FRPR-21, on hmc which inhibits hmc activation. Additional peptidergic input to hmc, possibly from AVL, contributes to hmc activation in some cycles.

distinct transmitters. Neurons that co-release both classical transmitters and neuropeptides are widespread. For example, in the spinal cord, the neuropeptides galanin, enkephalin or NPY are expressed in GABAergic neurons[50–52]. In *C. elegans*, the ASH sensory neuron secretes both fast neurotransmitters and neuropeptides to regulate multiple circuits for avoidance behaviors[53–55]. In bullfrogs, the co-release of luteinizing-hormone-releasing hormone and acetylcholine from spinal cord neurons controls the duration of post synaptic responses in C cells[56]. Single-cell expression profiling has revealed that neuropeptides are expressed in nearly every neuron in the mammalian cortex[57], and in *Drosophila*[58], indicating signaling by neuropeptides from multifunctional neurons will be an important way that information is possessed in the brain throughout phylogeny.

We speculate that FLP-22 is released from the proximal AVL axon, since this is the only part of AVL that is in direct contact with the pseudocoelom and that is in close proximity to hmc (Fig. 4a), and FLP-22 containing DCV clusters are found in the proximal AVL axons (Fig. S4a). In support of this idea, our live calcium imaging shows that AVL activates hmc rapidly (within 250 ms, Fig. 2b, c) following its activation, indicating that FLP-22 release must occur close to hmc to minimize the time of diffusion to its target receptors. In this case, FLP-22 would function as a short distance volume transmitter to activate hmc. Volume transmission over short distances by neuropeptides is widely observed. In mammalian KNDy neurons, kisspeptin functions as a volume transmitter to modulate the activity of neighboring GnRH neurons in the pituitary following its release at non-synaptic sites near the distal projections of GnRH neurons[11]. AVL may be an example of a neuron that exhibits spatially segregated release of its two transmitters, with FLP-22 released from extrasynaptic sites in the proximal axon and GABA released from NMJs at the tip. In *Aplysia*, the bag-cell peptide and the egg laying hormone, which control egg laying, are released from different projections of the bag cell neuron[59]. In the mammalian nervous system, there is evidence for spatially segregated release of at least two different transmitters from preganglionic sympathetic neurons[60], spinal motor neurons[61], and mesoaccumbens neurons[62]. The aBoc circuit is composed of at least two neuropeptide-like proteins that function as excitatory transmitters: NLP-40 released from the pacemaker depolarizes AVL (and DVB)[13,14], and FLP-22 released from AVL evokes a calcium transient in hmc. We propose that the precise timing information delivered by NLP-40 and FLP-22 for aBoc could be achieved by the rhythmic secretion of NLP-40 and FLP-22 in response to each calcium wave in the intestine and AVL, respectively. Rhythmic release of neuropeptides has been observed in regulating the circadian locomotor activity in *Drosophila*[63–65].

Our findings reveal that there is a network of additional cells and signals that may provide crosstalk, redundancy, and regulation to the main aBoc circuit (Fig. 6). First, although AVL is the primary source of FLP-22 for aBoc, only expression of *flp-22* in AVL plus additional GABAergic neurons fully rescues the aBoc defects of *flp-22* mutants (Fig. 1c), indicating that FLP-22 released from more than one source converges to activate hmc. Second, our calcium imaging indicates that FLP-22 may not be the only neuropeptide that activates hmc since in the absence of *flp-22* and *frpr-17*, hmc calcium spikes that trigger aBoc are still observed in 20% of cycles (Fig. 2f). Interestingly, the remaining hmc calcium spikes and aBocs in *flp-22* or *frpr-17* mutants are slightly delayed by about one second compared to wild-type controls (Fig. S5d), implying that the additional signal(s) may normally activate hmc slightly later than FLP-22. The use of multiple signals to activate hmc may explain the relatively long aBoc contraction duration (-1 s), and it may also ensure that a full aBoc occurs in every cycle by increasing the amplitude and/or duration of the calcium spike in hmc. The remaining hmc calcium spikes in *flp-22* or *frpr-17* mutants are larger in amplitude compared to wild-type controls (Fig. 2g), suggesting the existence of compensatory mechanisms that may either increase the strength of non-FLP-22 signaling or increase hmc

excitability in the absence of FLP-22. Third, we found that AVL ablation nearly eliminates aBoc (Fig. 1b), whereas elimination of hmc or *flp-22* signaling leads to absent aBoc in just about half of the cycles (Fig. 1d), indicating that AVL can control aBoc in an hmc- and *flp-22*-independent manner. One possibility is that the secretion of one or more non-FLP-22 peptides from AVL can bypass the requirement for hmc to promote aBoc. This signal is likely to be a neuropeptide since impairing neuropeptide processing (in *egl-3* mutants) leads to more severe aBoc defects than *flp-22* mutations (Fig. 1b). AVL expresses a number of neuropeptide-like proteins[26], that may be candidates for additional signals that control aBoc. The multiple layers of redundancy in this circuit may be an adaptation that ensures that the aBoc step remains robust so that animals can efficiently expel digested food under a variety of different environmental and/or physiological conditions.

Although hmc is born during embryogenesis, hmc does not appear to be essential for viability or for the development of the aBoc circuit (Fig. 1d). Instead, hmc plays a critical role in the mature aBoc circuit to couple *flp-22* signaling with neck muscle contraction. hmc expresses a number of muscle-specific genes (e.g. muscle myosins[25]), and has contractile fibers in the vicinity of the pharyngeal bulb[22], however, our analysis of *unc-54* and *unc-9* mutants suggests that its activation by *flp-22* signaling does not lead to its contraction (Fig. 4d, f, and Fig. S3c and Supplementary Movie 3 and 4), indicating that hmc is non-contractile for aBoc. Instead, hmc controls neck muscle contraction through the gap junctions that connect its dorsal and ventral processes to the dorsal and ventral neck muscles. Since FLP-22 is released from AVL on the ventral side of the animal, we speculate that FLP-22 likely activates FRPR-17 locally on the ventral hmc process nearest to the AVL axon, and the calcium wave is rapidly amplified and propagated throughout hmc, thereby ensuring isotonic hmc activation and the coordinated contraction of the dorsal and ventral neck muscles needed for aBoc. The neck muscles are also used for locomotion, and their activation out-of-phase with each other about once per second by motor neurons drives sinusoidal movement. Thus, the direct input to neck muscles provided by hmc may ensure that aBoc can be controlled independently of locomotion. A previous study found that locomotion and defecation are coordinated but this seems to occur upstream of AVL activation at the level of the intestine[20].

We found that PKA, which is activated by cAMP, functions in hmc to control aBoc frequency (Fig. 3). PKA dominant negative transgenes expressed in hmc nearly eliminate aBoc and significantly reduce hmc spike amplitude and duration. We speculate that FRPR-17 activates hmc primarily through PKA activation and that the remaining calcium spikes observed in animals expressing PKA dominant negative transgene reflect an incomplete knockout of PKA activity by the transgene. Nonetheless, the ability of PKA[DN] transgenes to eliminate aBoc and reduce calcium spike amplitude suggests that calcium spikes in hmc may need to rise above a threshold needed to trigger aBoc. An alternative explanation is that PKA is not the only effector of FRPR-17 in hmc. cAMP has multiple targets in addition to PKA including epac-1/EPAC or the cyclic nucleotide gated channels (*tax-2*, *tax-4*, *cng-1*, *cng-2*, and *cng-3*), but these genes are not expressed at detectable levels in hmc[25,26]. How does PKA activation lead to the generation of calcium transients? cAMP-PKA signaling has well established roles in controlling cell excitability. In the heart, PKA phosphorylates ryanodine receptors to release calcium from the sarcoplasmic reticulum (SR)[66], and PKA-mediated phosphorylation of L-type calcium channels leads to calcium influx[67]. In *C. elegans* motor neurons, PKA signaling regulates voltage-gated calcium channels to induce calcium influx in DVB[46]. hmc expresses all three VGCCs calcium channels (*unc-2*, *egl-19*, and *cca-1*) as well as *itr-1*/IP3 receptor and *unc-68*/ryanodine receptor at high levels[25]. These channels are also expressed in other tissues in the circuit, so addressing their role in hmc will require hmc-specific knock out strategies.

hmc is likely to activate the neck muscles exclusively through gap junctions, since eliminating *unc-9*/innexin causes similar reductions in aBoc frequency as ablating hmc (Figs. 1d and 4b). Our results suggest that the gap junctions that couple hmc to neck muscles are heterotypic. The hemichannels in hmc could be composed of either UNC-9/innexin homomers or heteromultimers in which UNC-9/innexin is an essential component. The hemichannels on the muscle side are likely to form heteromers composed of unidentified innexins that function redundantly since our screens did not identify additional innexins beyond *unc-9* that are required for aBoc. A number of heterotypic gap junctions have been reported, including between cardiac myocytes and surrounding fibroblasts[68,69], between germ cells and Sertoli cells in the testis[70], and between neuronal and glial cells in the nervous system[71]. In *C. elegans*, UNC-9 forms heterotypic gap junction with UNC-7 that couples AVB and B motor neurons for locomotion[72]. A recent study found that *frpr-17* signaling inhibits cAMP signaling to regulate gap junction assembly in motor neurons during development[73]. We do not believe that *frpr-17* functions in an analogous manner in hmc since the Gαs *gsa-1* is a positive regulator of aBoc and the hmc calcium spikes in *frpr-17* mutants are always accompanied by aBoc, implying that gap junctions are functional in these mutants. Since *frpr-17* mutants have defects in egg laying and locomotion not seen in *flp-22* mutants, we speculate that *frpr-17* may also function in these circuits using different ligands. Gap junctions have been reported to open and close and their opening can be regulated by PKA[74]. Whether gap junctions between hmc and neck muscles are gated is unclear. One observation making this idea favorable is that during live imaging we did not observe calcium transients in hmc when neck muscles contracted during locomotion suggesting that calcium generated in neck muscles for locomotion does not enter hmc.

The structural and functional characteristics of hmc resemble those of endothelial cells in vertebrates in a number of important respects. First, both cells are derived from mesoderm, are non-contractile, and are exposed to internal body cavities. Second, endothelial cells are exposed to the circulation and like hmc, the activation of GPCR signaling pathways by peptides in endothelial cells can induce changes in intracellular calcium levels[75]. Third, endothelial cells can form gap junctions with underlying smooth muscle and the communication between endothelial cells and smooth muscle cells via gap junctions can regulate the contraction state of smooth muscle cells[76]. Fourth, endothelial cells synthesize and release bioactive peptides[77], and hmc may also secrete peptides since expression profiling of hmc reveals high levels of a number of genes encoding signaling peptides, peptide processing enzymes, and peptide release machinery[25]. Finally, the *hlh-8*/TWIST transcription factor, which is important for the differentiation of hmc (Fig. 1d and[29]), is also expressed in endothelial cells, where it regulates endothelial cell proliferation and migration[78], consistent with a common origin of these cell types. However, terminal differentiation markers of endothelial cells are largely absent in *C. elegans*, and although both hmc and endothelial cells form a lining of the cavities they associate with, hmc does not form an epithelium. Given its long processes, hmc may be more similar in structure to amoebocytes in invertebrates, which are postulated to be ancestral endothelial cells of vertebrates[79].

## Methods

### Strains and transgenic lines

Strains were maintained at room temperature on nematode growth media (NGM) plates seeded with OP50 *Escherichia coli* as a food source. The wild-type strain was N2 Bristol. Transgenic lines were generated by injecting into adults with expression plasmids together with co-injection markers KP#708 (*Pttx-3::RFP* at 40 ng/μL) or KP#1338 (*Pttx-3::GFP* at 40 ng/μL) or KP#1106 (*Pmyo-2::NLS::GFP* at 5 ng/μL) or

KP#1368 (*Pmyo-2::NLS::mCherry* at 5 ng/μL) or pJQ70 (*Pofm-1::mCherry* at 25 ng/μL) or pDS806 (*Pmyo-3::mCherry* at 20 ng/μL) or pMH166 (*Plin-44::GFP* at 40 ng/μL). For tissue-specific gene expression, we used the following promoters: *Prab-3* for pan-neurons[80], *Punc-47* for GABAergic neurons[81], *Punc-129* for cholinergic neurons[82], *Pges-1* and *Pnlp-40* for intestine[14,83], and *Pmyo-3* for muscles[84]. Microinjection was conducted using standard procedures. In general, three lines were analyzed, and one representative line was used for quantification. The strains and transgenic lines used in this study are listed in Table S1.

### Molecular biology

All plasmids were constructed using the backbone of pPD49.26 or pPD95.75. Promoter regions were amplified from *C. elegans* genomic DNA and coding regions were amplified from mixed stage cDNA. PCR fragments were used to subclone into expression vectors using standard molecular biological techniques. mTur2 was subcloned from the pDD315 plasmid from Addgene https://www.addgene.org/73343/. A detailed list of plasmids and oligonucleotides used in this study are described in Table S1 and S2.

### *flp-22(OE)* suppressor screening and *frpr-17* cloning

The parental strain carrying the *Prab-3::flp-22* array (*flp-22(OE)*) was mutagenized with EMS for a standard non-clonal F2 screen. F2 progeny of ~10,000 mutagenized genomes were screened and one mutant that suppressed the uncoordinated phenotype of *flp-22(OE)* was identified. The *uj249* mutation was mapped to LG I using three point mapping. The lesion of *uj249* in the *frpr-17* gene was identified by whole genome sequencing (at the USC sequencing core) and the software MAQgene as previously described[85].

### CRISPR/Cas9-generated mutant strains

The deletion mutations in *flp-22*, *frpr-17*, *nmur-3*, *srd-32*, and *npr-23* were generated using a co-CRISPR strategy[86]. A sgRNA targeting *dpy-10* gene and a repair single-stranded oligodeoxynucleotides (ssODN), to generate a gain of function DPY-10, were co-injected with two sgRNA for gene of interest, one targeting around the first exon and other targeting around the last exon, and a ssODN to induce homology-directed repair (HDR). Fifteen adults were injected with the Cas9 enzyme containing mix of sgRNAs and ssODN. Twenty F1 animals were singled from five plates that carried Dpy or Rol phenotype. F2 animals were genotyped to select the deletion mutants. The homozygous mutants were outcrossed with wild-type at least four times before being used for experiments.

### Behavioral assays

The defecation motor program was analyzed as previously described[16,32]. Briefly, twenty to thirty L4 animals were moved to a fresh NGM plate seeded with OP50 bacterial lawn and were stored in a 20 °C incubator for 24 h. After 24 h, at least ten consecutive defecation cycles were observed from each animal under the dissecting microscope. The pBoc and aBoc steps were recorded using custom Etho software (James Thomas Lab website: http://depts.washington.edu/jtlab/software/otherSoftware.html). For egg laying assays, all eggs laid were counted from a single young adult animal over the course of four days. For analysis of locomotion activity, the number of body bends per 15 s was counted for young adult animals in the presence of food, as previously described[87]. At least three animals were assayed, and the mean and the standard error was calculated for each genotype.

### Feeding RNA interference

RNAi plates were made using established protocols. Seven gravid adult animals were bleached on RNAi pales seeded with HT115 (DE3) bacteria that was transformed with the targeted gene insert in the L4440 vector for knockdown. Three to four days later, adult animals were assayed for the defecation motor program. RNAi clones were from the

Ahringer or Vidal RNAi library. RNAi screening for neuropeptides was done in an *eri-1; lin-15* mutant background to increase RNAi efficacy in neurons[88]. RNAi for candidates functioning in hmc was done in an *eri-1* mutant background.

## Fluorescence imaging

Fluorescence imaging was done by using a Nikon eclipse 90i microscope equipped with a Nikon Plan Apo 20x, 40x, 60x, and 100x oil objective (N.A. = 1.40), and a Photometrics Coolsnap ES2 camera or a Hamamatsu Orca Flash LT + CMOS camera. L4 animals were transferred to a fresh NGM plate seeded with OP50 bacterial lawn and were stored in a 20 °C incubator for 24 h prior to imaging. Adult animals were paralyzed with 30 mg/ml 2, 3- Butanedione monoxime (BDM, Sigma) in M9 buffer, and then mounted on 2% agarose imaging pad. Metamorph 7.0 software (Universal Imaging) was used to capture image stacks and to obtain maximum intensity projections. All images were captured from left or right laterally positioned animals facing up. Fluorescence imaging of AVL and hmc were captured from the neck region where the terminal bulb of the pharynx is located. To analyze the ablation of hmc in N2 and *hlh-8*, GFP was expressed in both AVL and hmc (*Pnmur-3::GFP*) and the neck region was imaged in adult animals.

## Live calcium imaging

Calcium imaging was performed using freely moving adult animals. To limit the light stimulated movement while recording, *lite-3 gur-3* genetic background was introduced into wild-type and all the mutants used in this study[89,90]. Calcium imaging plates were prepared by making NGM plates with agarose instead of agar. OP50 bacteria were seeded on NGM-agarose plates 5 days prior to imaging. For calcium imaging simultaneously in the cell body of AVL and hmc, we used a transgenic line *vjEx2548* (*Pofm-1::mCherry*, *Pnmur-3::GCaMP6*, *Pnlp-40::GCaMP3*) to express GCaMP6 in AVL and hmc, and GCaMP3 in the intestine. Thirty to forty L4 stage animals were transferred to a normal NGM plate and were stored in a 20 °C incubator for 24 h prior to imaging. Adult animals were transferred to NGM-agarose plates seeded with OP50 and the plates were topped with a cover slide. Live imaging was done using a Nikon eclipse 90i microscope equipped with a Nikon Plan Apo 20× oil objective (N.A. = 1.0), a standard GFP filter and a Hamamatsu Orca Flash LT + CMOS camera. The animals that were pumping and positioned laterally with the left or right side facing the objective were selected for imaging. Metamorph 7.0 software (Universal Imaging) was used to obtain time lapse imaging. For each animal, the cell body of AVL and hmc together was recorded at 4 frames per second (2 × 2 binning with 30 ms exposure time).

The GCaMP6 fluorescence intensity in the cell body of AVL and hmc was quantified using ImageJ. The integrated fluorescence (F) of GCaMP6 was calculated by the integrated fluorescence density of a region of interest (ROI) within a 20 × 20 circle that covers the cell body of AVL or hmc minus the background integrated fluorescence density measured in the pharyngeal metacorpus. The baseline fluorescence ($F_0$) was defined by the average GCaMP6 fluorescence in the first 10 frames before the initiation of AVL or hmc activation. The fluorescent change of GCaMP6 for each frame was defined as $\Delta F/F_0 = (F-F_0)/F_0$. Only fluorescent changes of GCaMP in which the $\Delta F/F_0$ was greater than 50% of the baseline value were considered to be spikes. At least 5 cycles from at least 5 different animals were analyzed, and the mean and the standard error was calculated for each genotype.

aBocs were captured from the time lapse images during calcium imaging by observing the movement of the terminal pharyngeal bulb and measuring the area of the anterior intestinal lumen. In cycles with full aBoc, the terminal pharyngeal bulb rapidly displaces toward the anterior intestine, which is referred to as aBoc start, resulting in a shrinkage of the anterior intestinal lumen. The complete displacement of the anterior intestinal lumen is considered aBoc max, and

maintaining aBoc max for at least 500 ms is considered full aBoc. Partial aBocs refer to cycles with aBoc max lasting less than 500 ms. no aBocs refer to cycles with either no terminal pharyngeal bulb displacement or displacement that does not extend into the intestine.

## Cell ablation by miniSOG

Transgenic lines were generated by expressing membrane-targeted miniSOG in hmc (*Pnmur-3(Δ)::PH domain::miniSOG*). To ablate hmc, twenty to thirty L4 stage transgenic animals were transferred to an OP50 seeded NGM plate. The plate was illuminated with blue light using an EXFO mercury light source for 10 min with the cover off. Blue light illuminated animals were recovered at 20 °C for 24 h, and then assayed for the defecation motor program.

## Statistical analysis

Statistical analysis was performed using GraphPad Prism 9. All quantitative data was compared using two-tailed Student's *t* test (2 groups) or one-way ANOVA with multiple comparison corrections (3 or more groups) for parametric data. For non-parametric data, Kruskal-Wallis test with Dunn's correction for multiple comparisons was used. All categorical data was compared using two-sided Fisher's exact test (2 groups) or two-sided chi-square test with Bonferroni correction (3 or more groups). Statistical test, *P* values, and *n* are specified in the figure legends. All comparisons are to wild-type controls unless indicated by a line between genotypes.

## Reporting summary

Further information on research design is available in the Nature Portfolio Reporting Summary linked to this article.

## Data availability

All data that support the findings of this study are available within the paper and its supplementary files. Source data are provided with this paper. The source data can also be accessed at figshare. https://doi.org/10.6084/m9.figshare.22756898. Source data are provided with this paper.

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

## Acknowledgements

We thank members of the Sieburth lab for critical reading of the manuscript, to Han Wang for some reagents, and to Ariel Cohen for technical assistance. We thank Oliver Hobert for sharing the *hlh-8* mutant, and Kang Shen for sharing *GBD::sax-7* plasmids, and Laura Mathies for sharing RNAseq data with us, and David Hall for permission to include EM images. Some strains were provided by the CGC, which is funded by NIH Office of Research Infrastructure Programs (P40 OD010440), and the Japanese National Bioresource Project. The work is

funded by grants from National Institute of Health NINDS R01NS071085 and R01NS110730 (to D.S.).

## Author contributions
U.C., M.H., and D.S. conceived and designed research; U.C., M.H., and Q.Z. performed research; U.C., M.H., and D.S. contributed new reagents/analytic tools; U.C., M.H., and Q.Z. analyzed data; U.C. and D.S. wrote the paper.

## Competing interests
The authors declare no competing interests.
