## [Peer Review File · Nature Communications]

REVIEWER COMMENTS

Reviewer #1 (Remarks to the Author):

This study by Choi et al. explores how the hmc cell interfaces with the defecation motor circuit to regulate the aBOC step. The authors use extensive molecular genetic experiments to identify and show that FLP-22 neuropeptide is released from AVL and likely other GABA neurons that then signals through FRPR-17 to activate Gs and PKA signaling in hmc, modulating its calcium activity. The activity of hmc is also inhibited by FLP-9 which signals through FRPR-21 receptors. In addition, hmc activity is regulated by the UNC-9 innexin which may form gap junctions with anterior head muscles, allowing coordination of hmc and muscle activity during the aBOC step.

Overall, this is a very nice study, using a powerful combination of molecular genetics, forward and reverse genetics, calcium imaging to understand how several signals converge on an understudied but clearly excitable cell. Overall, the conclusions are well-supported by evidence presented, and I support publication, pending certain important revisions, as outlined below.

Major points:

1. After Figure 2B, I would prefer a 'summary' trace of calcium activity / behavior outlining each of the events, zoomed in on the actual aBoc. The authors make several conclusions on which events start when, with some cells responding to when peak activity is reached vs. the initial rise of activity, but that is very difficult to discern in this 150 s recording with different calcium levels trying to be compared. The authors should try to generate an 'average' version of this figure where calcium levels (hmc, AVL, and intestine) are normalized against each other and then those curves plotted against each other and aligned relative to a separate, objective behavior step (e.g. aBoc start / white box or aBoc max / black box, possibly placed at 0 seconds w/ ~5 seconds before and after). It could also be relative to pBoc if that's more 'objective'. I think that would help the readers understand the relative order of events more than graphs of spike initiation time do and description of the text every could. Of course, I only propose this summary analysis for the wild type.

2. Figure 2C presents a method of quantitation for 'spike' or 'no spike' but the criteria for this analysis is not explained. The authors must articulate how these events were called and whether they were done so objectively (e.g. if it was a threshold $\Delta F/F$ peak amplitude was used and what values were used for each cell).

3. Some of the statistical analyses performed appeared flawed to this reviewer. The methods states that Student's t tests were used for groups of 2 w/ ANOVA being done for greater than 3 w/ corrections for

multiple comparisons. While this is a good start, it is not enough information to interpret significance for the types of comparisons actually performed. For example, in the Figure 3A legend, the authors report p values from a Student's t test where multiple such comparisons are being shown (e.g. it's not just comparing two groups but multiple groups presumably against the same wild-type). It is not appropriate to use t-tests when two or much t-tests are being performed, the authors should use ANOVA. Is it possible the authors mis-stated the test and it was actually ANOVA? I have a feeling this isn't the case, though because of the spike/no-spike analyses where the authors report using the Fisher exact test which cannot be used to compare more than two outcomes / two conditions. This reviewer concludes that the authors did such pair-wise comparisons sequentially (not appropriate) without also providing evidence for how they then corrected for multiple comparisons. The Bonferroni correction can be used in such cases, but it is not an established feature in Prism and must be done manually. While I don't expect these issues to alter the conclusions of the paper materially, the authors must revisit their statistical procedures and make sure they are done correctly to ensure they are correcting for multiple comparisons whenever more than two such analyses are being performed on the same overall dataset. I believe this affects many/most Figures where the legends report p values from a t-test or a Fisher exact test. I believe the ANOVA analyses were performed and corrected appropriately.

Minor point:

The paper is written as if it is likely that hmc activity promotes the aBoc step, but most of the analysis could be interpreted the other way, that the AVL activates aBoc which then stimulates hmc. For example, it could be that AVL signals to enhance the excitability of hmc which then allows its mechanical activation in response to the aBoc. I think the only experiment that supports a conclusion directly that hmc activates aBoc is the unc-9 experiment where hmc still shows calcium activity in the absence of a (detectable) aBoc. I think the authors should take a little more care as they report their results that both possibilities are consistent with some of their genetic and calcium data, or to make clear the extent of the experimental evidence one way or the other. The normalized and averaged trace of the aBoc relative to AVL, hmc, and intestinal activity (Major point 1) might help support the authors conclusions that AVL then hmc activity precedes the aBoc and likely drives it via electrical signaling.

Reviewer #2 (Remarks to the Author):

In this paper, Choi and colleagues elucidate a cellular basis and molecular pathway in control of the anterior body wall muscle contraction (aBoc) during rhythmic defecation behavior in *C. elegans*. They identify a role for hmc, a cell of previously unknown function, in the defecation motor program and show by calcium imaging that it has rhythmic activity synchronized with the initiation of aBoc during defecation behavior. Using genetic ablation and loss-of-function analysis, Choi et al. characterize an

upstream peptidergic pathway, mediated by flp-22 signaling from AVL neurons, that controls hmc activity and aBoc cycles through the neuropeptide receptor FRPR-17. In addition, they demonstrate that hmc activation is facilitated by PKA signaling. Downstream of hmc activation, they find that the innexin UNC-9 is required for aBoc and may functionally couple hmc to the musculature via gap junctions. Finally, based on available gene expression data of hmc, they identify another neuromodulatory peptidergic pathway, mediated by flp-9 and frpr-21, that negatively regulates hmc activity.

The authors have presented a clear story and have used a broad range of genetic, ablation, calcium imaging, and behavioral analysis tools to support their results. The identification of a peptidergic and cellular mechanism underlying the defecation motor program is a valuable contribution to our understanding of behavioral rhythms and neuromodulation. The manuscript clearly describes the experiments, results and conclusions and is overall well-written. However, some experimental steps and statistical analyses remain unclear, and several concerns need to be addressed to substantiate the authors' conclusions.

Major comments:

1. When quantifying the number of aBoc per defecation cycle, statistical comparisons throughout the manuscript are done using a Student's t test regardless of the number of genotypes in the analysis (for example, in figures 1B-1D including data for up to 10 different genotypes). When analysing more than two strains, other statistical tests such as ANOVA are, however, more appropriate to avoid accumulation of type I errors. Throughout the manuscript, the use of Student's t tests should be reconsidered and corrected by the appropriate test where necessary. In addition, it is unclear from the methods and figure legends what the n numbers are in these experiments. The n numbers should be included in the figure legends and bar graphs should depict individual data points in addition to an average value and SEM.

2. For rescue experiments, it would be useful to include not only statistical comparisons to the mutant strains but also to the wild-type control, to indicate whether the transgene induces a full or partial rescue (for example, in figure 1C, hmc-specific rescue of frpr-17). In general, it is not always clear which reference strain has been used in a statistical comparison and the authors should try to clarify this further in the figure panels. For example, in Figure 5C, comparisons were made to wild type or to the frpr-17 single mutant? The hmc spike initiation of frpr-17 mutants does not differ from wild type in this analysis?

3. Use of the nmur-3 promoter in ablation and rescue experiments: According to single-cell RNAseq expression studies (Taylor et al., 2021), the nmur-3 promoter is also expressed in the DVB tail neuron, albeit at lower levels than in AVL. In figure S1, expression of different nmur-3 promoter fragments is shown in the head region. Did the authors also confirm that the AVL- and hmc-specific promoter

fragments were not expressed in DVB, or should possible effects of DVB be taken into account in ablation and rescue experiments?

4. To identify possible neuropeptides that control aBoc, an RNAi screen was performed using a feeding strategy. RNAi by feeding is known to be inefficient in eliciting a knockdown response in *C. elegans* neurons. Can the authors clarify their approach as well as potential measures they took to improve knockdown efficiency in the nervous system? In addition, it is curious that mutants of flp-22 still show 60% aBoc frequency, whereas RNAi-knockdown of flp-22 reduces aBoc to 25%. This suggests that there may be non-specific effects of the RNAi knockdown, potentially targeting other genes, which should be further addressed.

5. The authors state that FLP-22 neuropeptides act as direct transmitters signaling from AVL to hmc, and that AVL thus generates two divergent output signals – FLP-22 and GABA – to regulate two independent behaviors. This is supported by previous studies (e.g. Jin et al., 1999; Thomas, 1990) showing that unc-25 mutants lacking GABA have normal aBoc frequencies. However, it remains unclear whether GABAergic signaling is not involved in the regulation of hmc activity and whether unc-25 mutants may have subtle defects in the aBoc step (such as partial aBocs). To support this statement, hmc activity and the related aBoc cycles should be investigated in mutants deficient in GABAergic signaling from AVL.

6. Without additional experiments, the authors can only conclude that flp-22 and frpr-17 act in the same genetic pathway controlling aBoc. They cannot state that frpr-17 functions downstream of flp-22 in defecation behavior. This should be rephrased, or additional evidence should be provided. To identify potential FLP-22 receptors, they took advantage of the uncoordinated phenotype of a flp-22 overexpression strain to perform a suppressor screen. Do flp-22 OE strains also show aBoc defects? Can a potential ligand-receptor interaction between flp-22 and frpr-17 in the defecation motor program be further validated in vivo in this way?

7. Biochemical receptor activation studies have identified a number of G protein-coupled receptors that are activated by FLP-22 or FLP-9 peptides in vitro. These include DMSR-7 (FLP-22 and FLP-9) and EGL-6, FRPR-8 and DMSR-1 (FLP-9). However, it was not tested whether any of the FLP-22 or FLP-9 effects on aBoc or hmc activity could be mediated by these known receptors, potentially in addition to FRPR-17 and FRPR-21. These additional receptors should be included in the manuscript and a potential role of these GPCRs in the defecation circuit should be tested.

8. Calcium imaging experiments were performed using GCaMP sensors expressed specifically in the intestine, AVL soma or hmc cell body. These sensors were expressed from extrachromosomal arrays using tissue-specific promoters, which may show variable expression between individuals or in different

mutant backgrounds. To minimize artefacts with non-FRET indicators, dual color imaging of GCaMP and a calcium-independent red fluorescent signal is preferable and has been applied previously to AVL (e.g. by Jiang et al., Nat Comms, 2022). However, it is not clear which imaging strategy the authors used and how they dealt with potential artefacts, such as variation in sensor expression or motion artefacts during imaging of free-moving animals.

9. Previous work by Palumbos et al. (Dev Cell 2021) suggests that FRPR-17 inhibits cAMP signaling. This study finds that RNAi-mediated knockdown of *gsa-1* (G α -s) or *kin-1* (PKA catalytic subunit) reduces aBoc frequency and concludes that FRPR-17 controls aBoc by activating PKA in *hmc*. However, no clear evidence is presented for a genetic or biochemical interaction of FRPR-17 and the Gs pathway. Additional experiments are needed to support such an interaction, or the authors need to nuance this part of their working model. The PKA pathway could also regulate aBoc in response to other signals than FRPR-17 activation.

10. “*frpr-21* mutations significantly increased the calcium spike frequency in *frpr-17* mutants from 20% to 50% (Fig. 5B).” However, *frpr-21* mutations did not affect aBoc frequencies in *flp-22* mutants (Fig. 5A). Although these results indicate a role for FRPR-21 in regulating *hmc* calcium activity, it does not show a function in the control of rhythmic defecation behavior. Can the authors present additional evidence for such a role? For example, do the increased number of calcium spikes also increase the % aBoc in *frpr-17* mutants? To confirm a role of *flp-9* in the regulation of *hmc* calcium activity, the phenotype of *flp-9* mutants should also be rescued.

11. This study speculates on a model where the release of two signals – FLP-22 neuropeptides and GABA neurotransmitters – from AVL are differentially regulated in time, since the initiation of aBoc precedes the expulsion step by about 1 second. The authors state that “the calcium levels required for DCV release are significantly lower than those needed for SV release.” However, several studies indicate that the molecular pathways controlling DCV release are complex and diverse. In general, neuropeptide release is often thought to require high frequency stimulation of neurons and is not confined to synaptic or axonal locations. It is therefore unlikely that the calcium levels required for DCV release are consistently lower than those needed for SV release. Besides temporally regulated release, the time course of FLP-22 and GABA mediated events may also be explained by spatial factors, involving additional cells like the tail neuron DVB in the expulsion step. The authors should nuance their model in more detail, since there is no evidence supporting temporal release of neuropeptides and GABA from AVL neurons.

Minor comments:

1. Introduction on neuropeptide signaling: “Neuropeptides are commonly present as co-transmitters, but unlike fast neurotransmitters, neuropeptides act at slower timescales, can act over longer distances, and have longer lasting effects on target cell activity”. Please include references supporting these statements.

2. The authors list a number of examples of neuropeptide functions elucidated in *C. elegans*, but many more peptidergic systems have been functionally characterized in the nematode. They should indicate that these are some examples illustrating neuropeptide functions. The final conclusion of this paragraph “The functional significance of neuropeptide-mediated transmission in shaping behavioral outputs remains unknown” should also be more nuanced, as several studies have dissected the roles of specific peptidergic pathways in regulating behavioral outputs.

3. The authors use a series of cell- and tissue-specific promoters to drive targeted gene expression in the intestine (*ges-1*), GABAergic neurons (*unc-47*), cholinergic neurons (*unc-129*), etc. References to previous studies showing the cell- or tissue-specificity of these promoters should be included in this manuscript.

4. “Knockdown of *flp-22* by RNAi or knockout of *flp-22* by deleting the *flp-22* coding region significantly reduced aBoc frequencies to 60%.” A reference to figure 1C, including data of the *flp-22* mutant, is missing here.

5. “*flp-22* mutants had grossly normal locomotion and egg laying rates” and “a *frpr-17* null mutant that deletes the entire *frpr-17* coding sequence had aBoc frequencies of 60%, [...] and superficially normal locomotion and egg-laying”. No data on locomotion or egg laying is shown in the manuscript.

6. “Double mutants lacking both *frpr-17* and *flp-22* exhibited aBoc defects similar to those of either *flp-22* or *frpr-17* single mutants (Fig. 1C).” The statistical comparison between the double mutant and *flp-22* single mutant is not indicated in Fig. 1C. The authors also end this paragraph with the statement that “*frpr-17* functions downstream of *flp-22* in a genetic pathway to promote aBoc.” However, based on this double mutant analysis, it can only be stated that *flp-22* and *frpr-17* function in the same genetic pathway controlling aBoc, not that *frpr-17* acts downstream of *flp-22*.

7. In calcium imaging experiments, the time course of events is difficult to deduce from the representative images and traces presented in the figures. It would be useful to indicate the different time points and decay times (which are mentioned in the text) on the figure panels.

8. "The proximal axon of AVL lies in close proximity (within 5 uM) ..." Unit of distance is unclear or incorrect.

9. In the main text, the authors refer to SAX-7::GBD, whereas the images state GBP::SAX-7. Typo?

10. Fig. S4: "Animals expressing FLP-22::pHluorin in AVL exhibited very little fluorescence since pHluorin is quenched in DCVs, which are acidic." This control is not shown but should be included in the figure.

11. The authors locate the subcellular expression of a functional UNC-9::mTurq2 fusion protein, but what is the evidence that this fusion protein is indeed functional?

12. Page 14: "FRPR-17::Venus fusion proteins adopted a diffuse pattern of fluorescence in the hmc cell body and processes, consistent with surface expression throughout hmc (Fig. 5G)." Should be FRPR-21::Venus?

13. "frpr-21 null mutants exhibited aBoc frequencies and calcium spike frequencies in both AVL and hmc that were similar to wild-type controls (Fig. 5B and S6A), [...]" Also refer to Figure 5A here.

14. "flp-9 encodes a FMFR-like peptides [...]" Should be FMRFamide-like peptide.

15. "Overexpressing frpr-21 cDNA in hmc (frpr-21 (OE)) resulted in missing hmc calcium spikes in about 50% of cycles (Fig. 5F)." The figure shows missing calcium spikes in about 80% of cycles.

16. Discussion: typo Aplasia should be Aplysia.

17. Figure S1: "The Pnmur-3(Δ) promoter fragment extends from -2026 bp to -2956 bp and drives GFP expression in hmc but not in AVL. +++ indicates that 80-100% of animals exhibited fluorescence in the indicated cell." In how many animals was this examined?

18. Figs S3A + C: What are the n numbers?

REVIEWER COMMENTS

Reviewer #1 (Remarks to the Author):

This study by Choi et al. explores how the hmc cell interfaces with the defecation motor circuit to regulate the aBOC step. The authors use extensive molecular genetic experiments to identify and show that FLP-22 neuropeptide is released from AVL and likely other GABA neurons that then signals through FRPR-17 to activate Gs and PKA signaling in hmc, modulating its calcium activity. The activity of hmc is also inhibited by FLP-9 which signals through FRPR-21 receptors. In addition, hmc activity is regulated by the UNC-9 innexin which may form gap junctions with anterior head muscles, allowing coordination of hmc and muscle activity during the aBOC step.

Overall, this is a very nice study, using a powerful combination of molecular genetics, forward and reverse genetics, calcium imaging to understand how several signals converge on an understudied but clearly excitable cell. Overall, the conclusions are well-supported by evidence presented, and I support publication, pending certain important revisions, as outlined below.

Major points:

1. After Figure 2B, I would prefer a 'summary' trace of calcium activity / behavior outlining each of the events, zoomed in on the actual aBoc. The authors make several conclusions on which events start when, with some cells responding to when peak activity is reached vs. the initial rise of activity, but that is very difficult to discern in this 150 s recording with different calcium levels trying to be compared. The authors should try to generate an 'average' version of this figure where calcium levels (hmc, AVL, and intestine) are normalized against each other and then those curves plotted against each other and aligned relative to a separate, objective behavior step (e.g. aBoc start / white box or aBoc max / black box, possibly placed at 0 seconds w/ ~5 seconds before and after). It could also be relative to pBoc if that's more 'objective'. I think that would help the readers understand the relative order of events more than graphs of spike initiation time do and description of the text every could. Of course, I only propose this summary analysis for the wild type.

We have added a panel with a trace of average wild type calcium responses in AVL and hmc as well as the different phases of aBoc to Fig 2c. We normalized the peak intensities of hmc and AVL calcium responses, and indicated the initiation and maximum contractions of aBoc relative to the calcium traces. Since intestinal GCaMP intensity is dim and difficult to quantify due to intestinal background fluorescence, we used the initiation of the calcium spike in AVL as a reference point for all the events analyzed.

Page 7, Results:

The AVL and hmc calcium spikes initiated within 250ms of each other. AVL spikes peaked after about 1 s following spike initiation, averaging 2-fold increase from baseline, (Fig. 2) and decayed with an average half-decay time of 4.63 s (Fig. 2). hmc spikes were significantly larger in peak amplitude (averaging 12-fold increase from baseline) and decayed more slowly than those in AVL before returning to baseline with an average half-decay time of 9.55 s (Fig. 2). Each calcium spike was accompanied by an aBoc, which could be seen in the time-lapse images as a rapid posterior-directed displacement of the pharynx into the anterior intestine (Fig.

S3b). On average, aBocs initiated slightly after the initiation of the AVL and hmc calcium spikes and reached maximal contraction slightly before the peak of both spikes (Fig. 2c). Maximal contractions lasted for average 1.7s followed by a slower more variable relaxation lasting for a few seconds (Fig. 2c).

2. Figure 2C presents a method of quantitation for 'spike' or 'no spike' but the criteria for this analysis is not explained. The authors must articulate how these events were called and whether they were done so objectively (e.g. if it was a threshold $\Delta F/F$ peak amplitude was used and what values were used for each cell).

We have generated animals that co-express CGaMP and mCherry and measured fluorescence of both fluorophores in AVL and hmc during a cycle. We have added average mCherry fluorescence traces to Fig S3a. We have also added GCaMP fluorescence traces in hmc in *frpr-17* mutants to illustrate what 'no spike' looks like to Fig 2g. We added a description of how we determine a spike in the Methods.

Page 7, Results:

The GCaMP fluorescence spikes were not an artifact of movement or muscle contraction since in animals co-expressing both GCaMP and mCherry, mCherry fluorescence in AVL and hmc remained at baseline levels throughout the cycle, including during aBoc (Fig. S3a).

Page 8, Results:

frpr-17 mutants exhibited similar calcium phenotypes as *flp-22* mutants: calcium spike timing, frequency, peak amplitude and rise time in AVL were normal (Fig. S5a-c), but calcium spikes in hmc occurred in just 20% of cycles (Fig. 2f and Movie S2)...

Page 18, Methods:

Only fluorescent changes of GCaMP in which the $\Delta F/F_0$ was greater than 50% of the baseline value were considered to be spikes.

3. Some of the statistical analyses performed appeared flawed to this reviewer. The methods states that Student's t tests were used for groups of 2 w/ ANOVA being done for greater than 3 w/ corrections for multiple comparisons. While this is a good start, it is not enough information to interpret significance for the types of comparisons actually performed. For example, in the Figure 3A legend, the authors report p values from a Student's t test where multiple such comparisons are being shown (e.g. it's not just comparing two groups but multiple groups presumably against the same wild-type). It is not appropriate to use t-tests when two or much t-tests are being performed, the authors should use ANOVA. Is it possible the authors mis-stated the test and it was actually ANOVA? I have a feeling this isn't the case, though because of the spike/no-spike analyses where the authors report using the Fisher exact test which cannot be used to compare more than two outcomes / two conditions. This reviewer concludes that the authors did such pair-wise comparisons sequentially (not appropriate) without also providing evidence for how they then corrected for multiple comparisons. The Bonferroni correction can be used in such cases, but it is not an established feature in Prism and must be done manually. While I don't expect these issues to alter the conclusions of the paper materially, the authors must revisit their statistical procedures and make sure they are done correctly to ensure they are correcting for multiple comparisons whenever more than two such analyses are being performed on the same overall dataset. I

believe this affects many/most Figures where the legends report p values from a t-test or a Fisher exact test. I believe the ANOVA analyses were performed and corrected appropriately.

We have re-analyzed the statistics of quantitative data with ANOVA for any multiple comparisons. In addition, we have re-analyzed the statistics of spike/no-spike and aBoc frequencies from time lapses images using chi-square test with Bonferroni corrections for any graphs comparing more than two groups.

Page 18, Methods:

Statistical analysis was performed using GraphPad Prism 9. All quantitative data was compared using Student's t test (2 groups) or one-way ANOVA with multiple comparison corrections (3 or more groups) for parametric data. For non-parametric data, Kruskal-Wallis test with Dunn's correction for multiple comparisons was used. All categorical data was compared using Fisher's exact test (2 groups) or Chi-square test with Bonferroni correction (3 or more groups). Statistical test, P values, and N are specified in the figure legends. All comparisons are compared to wild-type controls unless indicated by a line between genotypes.

Minor point:

The paper is written as if it is likely that hmc activity promotes the aBoc step, but most of the analysis could be interpreted the other way, that the AVL activates aBoc which then stimulates hmc. For example, it could be that AVL signals to enhance the excitability of hmc which then allows its mechanical activation in response to the aBoc. I think the only experiment that supports a conclusion directly that hmc activates aBoc is the *unc-9* experiment where hmc still shows calcium activity in the absence of a (detectable) aBoc. I think the authors should take a little more care as they report their results that both possibilities are consistent with some of their genetic and calcium data, or to make clear the extent of the experimental evidence one way or the other. The normalized and averaged trace of the aBoc relative to AVL, hmc, and intestinal activity (Major point 1) might help support the authors conclusions that AVL then hmc activity precedes the aBoc and likely drives it via electrical signaling.

We agree that the *unc-9* data supports the conclusion that hmc activates muscle and not vice versa. We have added an additional supporting experiment using *unc-54*/myosin mutants, which have defects in muscle contraction, to Fig S3c and d. We find that in cycles without aBoc, AVL and hmc are still normally activated in these mutants, showing that muscle contraction is not necessary for hmc activation.

Page 8, Results:

To test whether hmc activation controls neck muscle contraction or vice versa, we examined mutants in *unc-54*, which encodes myosin heavy chain and is required for calcium-dependent contraction of body wall muscles (MacLeod et al., 1981). As expected, *unc-54* loss-of-function mutants had no detectable or very weak aBocs (Fig. S3c), yet we always observed calcium responses in hmc that were similar in frequency and amplitude to those of wild type controls (Fig. S3d, Movie S4). Thus, hmc can be activated even when neck muscle contraction is severely compromised, indicating that the neck muscles are unlikely to activate hmc. Together, we conclude that AVL is upstream of hmc and hmc is upstream of neck muscle in the aBoc circuit (Fig. 1a).

Reviewer #2 (Remarks to the Author):

In this paper, Choi and colleagues elucidate a cellular basis and molecular pathway in control of the anterior body wall muscle contraction (aBoc) during rhythmic defecation behavior in *C. elegans*. They identify a role for hmc, a cell of previously unknown function, in the defecation motor program and show by calcium imaging that it has rhythmic activity synchronized with the initiation of aBoc during defecation behavior. Using genetic ablation and loss-of-function analysis, Choi et al. characterize an upstream peptidergic pathway, mediated by flp-22 signaling from AVL neurons, that controls hmc activity and aBoc cycles through the neuropeptide receptor FRPR-17. In addition, they demonstrate that hmc activation is facilitated by PKA signaling. Downstream of hmc activation, they find that the innexin UNC-9 is required for aBoc and may functionally couple hmc to the musculature via gap junctions. Finally, based on available gene expression data of hmc, they identify another neuromodulatory peptidergic pathway, mediated by flp-9 and frpr-21, that negatively regulates hmc activity.

The authors have presented a clear story and have used a broad range of genetic, ablation, calcium imaging, and behavioral analysis tools to support their results. The identification of a peptidergic and cellular mechanism underlying the defecation motor program is a valuable contribution to our understanding of behavioral rhythms and neuromodulation. The manuscript clearly describes the experiments, results and conclusions and is overall well-written. However, some experimental steps and statistical analyses remain unclear, and several concerns need to be addressed to substantiate the authors' conclusions.

Major comments:

1. When quantifying the number of aBoc per defecation cycle, statistical comparisons throughout the manuscript are done using a Student's t test regardless of the number of genotypes in the analysis (for example, in figures 1B-1D including data for up to 10 different genotypes). When analysing more than two strains, other statistical tests such as ANOVA are, however, more appropriate to avoid accumulation of type I errors. Throughout the manuscript, the use of Student's t tests should be reconsidered and corrected by the appropriate test where necessary. In addition, it is unclear from the methods and figure legends what the n numbers are in these experiments. The n numbers should be included in the figure legends and bar graphs should depict individual data points in addition to an average value and SEM.

We have re-analyzed the statistics of quantitative data with ANOVA for any multiple comparisons. In addition, we have re-analyzed the statistics of spike/no-spike and aBoc frequencies from time lapses images using chi-square test with Bonferroni corrections for any graphs comparing more than two groups. We have provided the n numbers in the methods and depicted the individual data points on graphs for all the quantitative data.

Page 18, Methods:

Statistical analysis

Statistical analysis was performed using GraphPad Prism 9. All quantitative data was compared using Student's t test (2 groups) or one-way ANOVA with multiple comparison corrections (3 or

more groups) for parametric data. For non-parametric data, Kruskal-Wallis test with Dunn's correction for multiple comparisons was used. All categorical data was compared using Fisher's exact test (2 groups) or Chi-square test with Bonferroni correction (3 or more groups). Statistical test, *P* values, and *N* are specified in the figure legends.

Page 17, Methods for aBoc assay:

At least three animals were assayed, and the mean and the standard error was calculated for each genotype.

Page 18, Methods for live calcium imaging:

At least 5 cycles from at least 5 different animals were analyzed, and the mean and the standard error was calculated for each genotype.

2. For rescue experiments, it would be useful to include not only statistical comparisons to the mutant strains but also to the wild-type control, to indicate whether the transgene induces a full or partial rescue (for example, in figure 1C, hmc-specific rescue of *frpr-17*). In general, it is not always clear which reference strain has been used in a statistical comparison and the authors should try to clarify this further in the figure panels. For example, in Figure 5C, comparisons were made to wild type or to the *frpr-17* single mutant? The hmc spike initiation of *frpr-17* mutants does not differ from wild type in this analysis?

We have indicated whether comparisons were made to wild type or to mutants using horizontal lines in the figures throughout.

Page 18, Methods:

All comparisons are compared to wild-type controls unless indicated by a line between genotypes.

3. Use of the *nmur-3* promoter in ablation and rescue experiments: According to single-cell RNAseq expression studies (Taylor et al., 2021), the *nmur-3* promoter is also expressed in the DVB tail neuron, albeit at lower levels than in AVL. In figure S1, expression of different *nmur-3* promoter fragments is shown in the head region. Did the authors also confirm that the AVL- and hmc-specific promoter fragments were not expressed in DVB, or should possible effects of DVB be taken into account in ablation and rescue experiments?

We have modified Fig S1 to include expression information for this promoter in DVB. We see no expression in DVB in the hmc-specific promoter fragment and we see weak DVB expression in 10% of animals in the AVL-specific promoter fragment.

Page 31, Figure S1a legend:

(a) Schematic of the *nmur-3* promoter. The *nmur-3(3kb)* promoter fragment extends from -1bp to -2956bp relative to the ATG codon of *nmur-3*, and drives expression of GFP in AVL, DVB and hmc. The *Pnmur-3(1kb)* promoter fragment extends from -1bp to -1053 bp relative to the ATG codon of *nmur-3*, and drives GFP expression in AVL but not in hmc. The *Pnmur-3(Δ)* promoter

fragment extends from -2026 bp to -2956 bp and drives GFP expression primarily in hmc but not in AVL. +++ indicates that 80-100% of animals exhibited fluorescence in the indicated cell, + indicates that <10% of animals exhibited fluorescence in the indicated cell, and – indicates that 0% of animals exhibited fluorescence in the indicated cell. At least 20 animals were examined for each transgenic line.

4. To identify possible neuropeptides that control aBoc, an RNAi screen was performed using a feeding strategy. RNAi by feeding is known to be inefficient in eliciting a knockdown response in *C. elegans* neurons. Can the authors clarify their approach as well as potential measures they took to improve knockdown efficiency in the nervous system? In addition, it is curious that mutants of flp-22 still show 60% aBoc frequency, whereas RNAi-knockdown of flp-22 reduces aBoc to 25%. This suggests that there may be non-specific effects of the RNAi knockdown, potentially targeting other genes, which should be further addressed.

We have modified the methods to clarify that genetic background that the RNAi experiments were done in was *eri-1; lin-15b*. The control *eri-1; lin-15b* strain has slightly lower aBoc frequency than N2, possibly explaining why RNAi of flp-22 caused a more severe defect than the mutant.

Page 17, Methods:

RNAi screening for neuropeptides was done in an *eri-1; lin-15* mutant background to increase RNAi efficacy in neurons (Wang et al., 2005).

5. The authors state that FLP-22 neuropeptides act as direct transmitters signaling from AVL to hmc, and that AVL thus generates two divergent output signals – FLP-22 and GABA – to regulate two independent behaviors. This is supported by previous studies (e.g. Jin et al., 1999; Thomas, 1990) showing that *unc-25* mutants lacking GABA have normal aBoc frequencies. However, it remains unclear whether GABAergic signaling is not involved in the regulation of hmc activity and whether *unc-25* mutants may have subtle defects in the aBoc step (such as partial aBocs). To support this statement, hmc activity and the related aBoc cycles should be investigated in mutants deficient in GABAergic signaling from AVL.

We have added *unc-25* imaging analysis and aBoc analysis to Fig S7d.

Page 11, Results:

Similarly, we found that *unc-25*/glutamate decarboxylase mutants exhibited similar calcium spike frequencies in AVL and hmc as wild-type controls and did not alter hmc activation or aBoc frequency of *frpr-17* mutants (Fig. S7d).

6. Without additional experiments, the authors can only conclude that flp-22 and frpr-17 act in the same genetic pathway controlling aBoc. They cannot state that frpr-17 functions downstream of flp-22 in defecation behavior. This should be rephrased, or additional evidence should be provided. To identify potential FLP-22 receptors, they took advantage of the uncoordinated phenotype of a flp-22 overexpression strain to perform a suppressor screen. Do flp-22 OE strains also show aBoc defects? Can a potential ligand-receptor interaction between flp-22 and frpr-17 in the defecation motor program be further validated in vivo in this way?

Flp-22 overexpressing strains have only mild aBoc defects, so they were not useful for addressing ligand-receptor interaction. We removed strong conclusions regarding that *frpr-17* functions downstream of *flp-22* from the Results. We still include this idea in our model, since it is consistent with our data.

7. Biochemical receptor activation studies have identified a number of G protein-coupled receptors that are activated by FLP-22 or FLP-9 peptides in vitro. These include DMSR-7 (FLP-22 and FLP-9) and EGL-6, FRPR-8 and DMSR-1 (FLP-9). However, it was not tested whether any of the FLP-22 or FLP-9 effects on aBoc or hmc activity could be mediated by these known receptors, potentially in addition to FRPR-17 and FRPR-21. These additional receptors should be included in the manuscript and a potential role of these GPCRs in the defecation circuit should be tested.

We have added calcium imaging for *dmsr-1*, *dmsr-7*, *frpr-8* and *egl-6* mutants to Fig S8d.

Page 11, Results:

FLP-9 is reported to interact with four neuropeptide GPCRs in in vitro binding assays (DMSR-1, DMSR-7, EGL-6 and FRPR-8), and FLP-22 is reported to interact with DMSR-7 (Beets et al., 2022). Putative null mutants in each of these GPCRs did not alter hmc activation frequency in wild-type or in *frpr-17* mutant backgrounds (Fig. S8d), suggesting that FLP-9 and FLP-22 are unlikely to control hmc activation via these GPCRs.

8. Calcium imaging experiments were performed using GCaMP sensors expressed specifically in the intestine, AVL soma or hmc cell body. These sensors were expressed from extrachromosomal arrays using tissue-specific promoters, which may show variable expression between individuals or in different mutant backgrounds. To minimize artefacts with non-FRET indicators, dual color imaging of GCaMP and a calcium-independent red fluorescent signal is preferable and has been applied previously to AVL (e.g. by Jiang et al., Nat Comms, 2022). However, it is not clear which imaging strategy the authors used and how they dealt with potential artefacts, such as variation in sensor expression or motion artefacts during imaging of free-moving animals.

We have generated animals co-expressing GCaMP and mCherry in AVL and hmc and performed live calcium imaging. We found that GCaMP fluorescence spikes during aBoc, whereas there is very little detectable change in mCherry fluorescence. This data had been added to Fig S3a.

Page 7, Results:

The GCaMP fluorescence spikes were not an artifact of movement or muscle contraction since in animals co-expressing both GCaMP and mCherry, mCherry fluorescence in AVL and hmc remained at baseline levels throughout the cycle, including during aBoc (Fig. S3a).

9. Previous work by Palumbos et al. (Dev Cell 2021) suggests that FRPR-17 inhibits cAMP signaling. This study finds that RNAi-mediated knockdown of *gsa-1* (Galpha-s) or *kin-1* (PKA catalytic subunit) reduces aBoc frequency and concludes that FRPR-17 controls aBoc by activating PKA in hmc. However, no clear evidence is presented for a genetic or biochemical interaction of FRPR-17 and the Gs pathway. Additional experiments are needed to support such an interaction, or the authors need to nuance this part of their working model. The PKA pathway could also regulate aBoc in response to other signals than FRPR-17 activation.

We agree and we have toned this conclusion down. We have also referenced this work in the Discussion.

Page 15, Discussion:

A recent study found that *frpr-17* signaling inhibits cAMP signaling to regulate gap junction assembly in motor neurons during development (Palumbos et al., 2021). We do not believe that *frpr-17* functions in an analogous manner in hmc since *gsa-1* is a positive regulator of aBoc and the hmc calcium spikes in *frpr-17* mutants are always accompanied by aBoc, implying that gap junctions are functional in these mutants.

10. “*frpr-21* mutations significantly increased the calcium spike frequency in *frpr-17* mutants from 20% to 50% (Fig. 5B).” However, *frpr-21* mutations did not affect aBoc frequencies in *flp-22* mutants (Fig. 5A). Although these results indicate a role for FRPR-21 in regulating hmc calcium activity, it does not show a function in the control of rhythmic defecation behavior. Can the authors present additional evidence for such a role? For example, do the increased number of calcium spikes also increase the % aBoc in *frpr-17* mutants? To confirm a role of *flp-9* in the regulation of hmc calcium activity, the phenotype of *flp-9* mutants should also be rescued.

We have added *flp-9* rescue data using its endogenous promoter fig 5b. We found that *flp-9* mutations cause a small but not significant increase in aBoc frequency in *frpr-17* mutants (Fig 5a).

Page 11, Results:

Expression of *flp-9* cDNA under an endogenous promoter fragment (*Pflp-9::flp-9*) fully reverted the hmc activation phenotype of *flp-9; frpr-17* double mutants to 20% (Fig. 5b).

11. This study speculates on a model where the release of two signals – FLP-22 neuropeptides and GABA neurotransmitters – from AVL are differentially regulated in time, since the initiation of aBoc precedes the expulsion step by about 1 second. The authors state that “the calcium levels required for DCV release are significantly lower than those needed for SV release.” However, several studies indicate that the molecular pathways controlling DCV release are complex and diverse. In general, neuropeptide release is often thought to require high frequency stimulation of neurons and is not confined to synaptic or axonal locations. It is therefore unlikely that the calcium levels required for DCV release are consistently lower than those needed for SV release. Besides temporally regulated release, the time course of FLP-22 and GABA mediated events may also be explained by spatial factors, involving additional cells like the tail neuron DVB in the expulsion step. The authors should nuance their model in more detail, since there is no evidence supporting temporal release of neuropeptides and GABA from AVL neurons.

We agree that ideas about timing are speculative, so we have eliminated this paragraph from the Discussion.

Minor comments:

1. Introduction on neuropeptide signaling: “Neuropeptides are commonly present as co-transmitters,

but unlike fast neurotransmitters, neuropeptides act at slower timescales, can act over longer distances, and have longer lasting effects on target cell activity". Please include references supporting these statements.

We have added references and we have added examples of neuropeptides functioning as transmitters to the introduction.

Page 3, Introduction:

Neuropeptides are commonly present as co-transmitters, but unlike fast neurotransmitters, neuropeptides act at slower timescales, can act over long distances, and have longer lasting effects on target cell activity (Bhat et al., 2021; Taghert and Veenstra, 2003). Neuropeptide signaling in the brain has been typically regarded as modulatory, whereby neuropeptides positively or negatively regulate postsynaptic responses elicited by fast neurotransmitters (Taghert and Nitabach, 2012). However, neuropeptides can also function as excitatory transmitters in their own right by directly activating targets cells. In circuits regulating wakefulness, orexin secreted from the lateral hypothalamus generates robust postsynaptic spike trains in target neurons that are independent of those produced by co-released glutamate (Schone et al., 2014). Pulse-generating KDNy cells in the hypothalamus control episodic activation of target neurons through the release of the FMRFamide-like neuropeptide kisspeptin (Liu et al., 2021). The neuropeptide FMRFamide elicits fast depolarizing inward currents in the snail nervous system (Cottrell et al., 1990). Finally, in *C. elegans* the neuropeptide-like protein NLP-40 depolarizes a pair of GABAergic motor neurons to control a rhythmic behavior (Jiang et al., 2022; Wang et al., 2013).

2. The authors list a number of examples of neuropeptide functions elucidated in *C. elegans*, but many more peptidergic systems have been functionally characterized in the nematode. They should indicate that these are some examples illustrating neuropeptide functions. The final conclusion of this paragraph "The functional significance of neuropeptide-mediated transmission in shaping behavioral outputs remains unknown" should also be more nuanced, as several studies have dissected the roles of specific peptidergic pathways in regulating behavioral outputs.

We have removed this sentence.

3. The authors use a series of cell- and tissue-specific promoters to drive targeted gene expression in the intestine (*ges-1*), GABAergic neurons (*unc-47*), cholinergic neurons (*unc-129*), etc. References to previous studies showing the cell- or tissue-specificity of these promoters should be included in this manuscript.

We have added these references.

Page 16, Methods:

For tissue-specific gene expression, we used the following promoters: *Prab-3* for pan-neurons (Nonet et al., 1997), *Punc-47* for GABAergic neurons (McIntire et al., 1997), *Punc-129* for cholinergic neurons (Sieburth et al., 2005), *Pges-1* and *Pnlp-40* for intestine (Egan et al., 1995; Wang et al., 2013), and *Pmyo-3* for muscles (Ardizzi and Epstein, 1987).

4. “Knockdown of flp-22 by RNAi or knockout of flp-22 by deleting the flp-22 coding region significantly reduced aBoc frequencies to 60%.” A reference to figure 1C, including data of the flp-22 mutant, is missing here.

Added.

5. “flp-22 mutants had grossly normal locomotion and egg laying rates” and “a frpr-17 null mutant that deletes the entire frpr-17 coding sequence had aBoc frequencies of 60%, [...] and superficially normal locomotion and egg-laying”. No data on locomotion or egg laying is shown in the manuscript.

We have added egg laying and locomotion data to Fig S2 e and f.

Page 5, Results:

flp-22 mutants exhibited locomotion rates and egg laying rates that were similar to wild type controls (Fig. S2e, f).

Page 6, Results:

Locomotion and egg laying rates of *frpr-17* mutants were reduced compared to wild type controls (Fig. S2e, f).

6. “Double mutants lacking both frpr-17 and flp-22 exhibited aBoc defects similar to those of either flp-22 or frpr-17 single mutants (Fig. 1C).” The statistical comparison between the double mutant and flp-22 single mutant is not indicated in Fig. 1C. The authors also end this paragraph with the statement that “frpr-17 functions downstream of flp-22 in a genetic pathway to promote aBoc.” However, based on this double mutant analysis, it can only be stated that flp-22 and frpr-17 function in the same genetic pathway controlling aBoc, not that frpr-17 acts downstream of flp-22.

We have modified this conclusion and added the statistical comparison.

7. In calcium imaging experiments, the time course of events is difficult to deduce from the representative images and traces presented in the figures. It would be useful to indicate the different time points and decay times (which are mentioned in the text) on the figure panels.

We have added a panel to figure 2c showing the time course of AVL activation, hmc activation and aBoc in more detail.

8. “The proximal axon of AVL lies in close proximity (within 5 μ M) ...” Unit of distance is unclear or incorrect.

Corrected.

9. In the main text, the authors refer to SAX-7::GBD, whereas the images state GBP::SAX-7. Typo?

Corrected.

10. Fig. S4: “Animals expressing FLP-22::pHluorin in AVL exhibited very little fluorescence since pHluorin is quenched in DCVs, which are acidic.” This control is not shown but should be included in the figure.

We have included an *unc-32* control to figure S4a

Page 8, Results:

pHluorin is quenched in DCV lumens due to their acidity, but pHluorin is unquenched in mutants lacking *unc-32*, which encodes the V_0 subunit of the vesicular ATPase proton transporter responsible for acidifying DCVs (Paquin et al., 2016). Fluorescence was not visible in AVL axons of wild type animals expressing FLP-22::pHluorin, but a highly punctate pattern of fluorescence characteristic of DCVs was visible in the proximal AVL axon of *unc-32* mutants (Fig. S4a), indicating that FLP-22 is packaged into DCVs in the proximal AVL axon.

11. The authors locate the subcellular expression of a functional UNC-9::mTurq2 fusion protein, but what is the evidence that this fusion protein is indeed functional?

In Fig 4b we show the results of rescue of *unc-9* mutants with *unc-9::mTur2* fusion proteins.

Page 26, Figure 4:

(b) Quantification of the number of aBocs per cycle in adult animals of the indicated genotypes. “*hmc unc-9*” and “*muscle unc-9*” denote expressing *unc-9::mTur2* fusion proteins or *unc-9* cDNA under the *nmur-3(Δ)* and *myo-3* promoter, respectively. Means and standard errors are shown. *** $P < 0.001$ and ** $P < 0.01$ in ANOVA with Bonferroni’s correction for multiple comparisons; n.s., not significant.

12. Page 14: “FRPR-17::Venus fusion proteins adopted a diffuse pattern of fluorescence in the hmc cell body and processes, consistent with surface expression throughout hmc (Fig. 5G).” Should be FRPR-21::Venus?

Corrected.

13. “*frpr-21* null mutants exhibited aBoc frequencies and calcium spike frequencies in both AVL and hmc that were similar to wild-type controls (Fig. 5B and S6A), [...]” Also refer to Figure 5A here.

Corrected.

14. “*flp-9* encodes a FMFR-like peptides [...]”. Should be FMRFamide-like peptide.

Corrected.

15. “Overexpressing *frpr-21* cDNA in hmc (*frpr-21* (OE)) resulted in missing hmc calcium spikes in about 50% of cycles (Fig. 5F).” The figure shows missing calcium spikes in about 80% of cycles.

Corrected.

16. Discussion: typo Aplasia should be Aplysia.

Corrected.

17. Figure S1: “The Pnmur-3(Δ) promoter fragment extends from -2026 bp to -2956 bp and drives GFP expression in hmc but not in AVL. +++ indicates that 80-100% of animals exhibited fluorescence in the indicated cell.” In how many animals was this examined?

We have added in the figure legend.

Page 31, Figure S1:

At least 20 animals were examined for each transgenic line.

18. Figs S3A + C: What are the n numbers?

We have updated all graphs throughout the manuscript to include the individual data points

Page 37, Figure S5:

For a, b, and d, wild-type: 33 cycles in 8 animals, *flp-22*: 49 cycles in 9 animals, *frpr-17*: 56 cycles in 13 animals, *flp-22; frpr-17*: 43 cycles in 9 animals, *frpr-17; hmc frpr-17*: 20 cycles in 7 animals, *flp-22; frpr-17; hmc frpr-17*: 42 cycles in 9 animals.

REVIEWERS' COMMENTS

Reviewer #1 (Remarks to the Author):

My issues have been addressed. Congratulations on a very nice study.

Reviewer #2 (Remarks to the Author):

The authors have performed a great amount of work in response to the reviewers' comments, including additional experiments and analyses, and content revisions. This additional work has substantially improved the quality of the manuscript and the authors have satisfactorily addressed all previous comments.

Two minor points may have been overlooked in the revised version:

1. Lines 391-392: "frpr-21;frpr-17 double mutants had slightly higher aBoc frequency compared to frpr-17 mutants, but this increase did not reach significance (Fig. 5a)." The revised figure 5a shows aBoc frequency for frpr-21;flp-22 double mutants but not for frpr-21;frpr-17 mutants or frpr-17 mutants. Is this sentence or figure reference correct?
2. Fig. 1c: The authors mentioned that a statistical comparison between the frpr-17;flp-22 double mutant and flp-22 single mutants was added in the revised manuscript. The revised Fig. 1c panel has a comparison between frpr-17;flp-22 and frpr-17 single mutants, but not flp-22 single mutants.

REVIEWERS' COMMENTS

Reviewer #1 (Remarks to the Author):

My issues have been addressed. Congratulations on a very nice study.

Reviewer #2 (Remarks to the Author):

The authors have performed a great amount of work in response to the reviewers' comments, including additional experiments and analyses, and content revisions. This additional work has substantially improved the quality of the manuscript and the authors have satisfactorily addressed all previous comments.

Two minor points may have been overlooked in the revised version:

1. Lines 391-392: "frpr-21;frpr-17 double mutants had slightly higher aBoc frequency compared to frpr-17 mutants, but this increase did not reach significance (Fig. 5a)." The revised figure 5a shows aBoc frequency for frpr-21;flp-22 double mutants but not for frpr-21;frpr-17 mutants or frpr-17 mutants. Is this sentence or figure reference correct?

We have corrected the statement in the results.

Page 11, results:

flp-22; frpr-21 double mutants had slightly higher aBoc frequency compared to *flp-22* mutants, but this increase did not reach significance (Fig. 5a).

2. Fig. 1c: The authors mentioned that a statistical comparison between the frpr-17;flp-22 double mutant and flp-22 single mutants was added in the revised manuscript. The revised Fig. 1c panel has a comparison between frpr-17;flp-22 and frpr-17 single mutants, but not flp-22 single mutants.

We have added P values between flp-22 single and flp-22; frpr-17 double mutants in Fig. 1c.